# What Does Flow-Matching Bring to TD-Learning?

**Bhavya Agrawalla** [1]   **Michal Nauman** [2]   **Aviral Kumar** [1]

## Abstract

Recent work shows that flow-matching networks can be effective for value function estimation in reinforcement learning, but it remains unclear why they work well or whether flow-matching Q-functions differ fundamentally from standard critics. We show that their success is not explained by distributional RL: explicitly modeling return distributions often degrades performance. Instead, we argue that flow-matching Q-functions are effective because they couple a learned velocity field with an integration procedure that is used both during training and to read out Q-values at inference time. This coupling enables robust value prediction through *test-time recovery* from imperfect intermediate estimates where errors dampen out as more integration steps are performed. This mechanism is absent in monolithic critics. Beyond test-time recovery, training with the integration procedure induces more *plastic* representations, allowing critics to represent non-stationary future TD targets without overwriting previous features. We formalize these effects and validate them empirically, showing that flow-matching critics outperform monolithic critics by over $2\times$ in performance and achieve $5$–$10\times$ higher sample efficiency in high-UTD regimes.

## 1. Introduction

Recent works have demonstrated that flow-matching networks can be highly effective for value function estimation in off-policy reinforcement learning (RL) (Agrawalla et al., 2025; Espinosa-Dice et al., 2025a; Dong et al., 2025). These flow-matching critics depart from standard "monolithic" architectures, which map state-action pairs to scalar Q-values in a single forward pass, by instead estimating values via the iterative integration of a learned velocity field given a noise input. This approach yields substantial empirical gains, and

[1]Computer Science Department, Carnegie Mellon University, USA [2]University of Warsaw, Poland. Correspondence to: Bhavya Agrawalla <bbagrawa@andrew.cmu.edu>.

*Proceedings of the 43rd International Conference on Machine Learning*, Seoul, South Korea. PMLR 306, 2026. Copyright 2026 by the author(s).

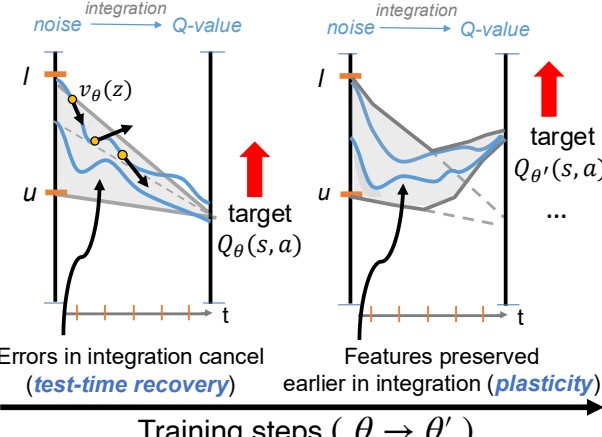

*Figure 1.* ***Flow-matching critics under non-stationary TD targets.*** As training proceeds ($\theta \to \theta'$), flow-matching critics adapt through a combination of integration ($t$) and updating the weights of the velocity network $v_\theta$ ($\theta \to \theta'$), resulting in **plastic** features and **test-time recovery** from imperfect value estimates.

is more robust than monolithic networks in offline (Levine et al., 2020) and offline-to-online RL settings (Nakamoto et al., 2024). However, while the performance gap is clear, the mechanism driving it is unexplained: do flow-matching critics succeed because they model a value distribution, or do they offer a different inductive bias that limits the pathologies of temporal-difference (TD) learning?

A natural hypothesis is that flow-matching critics succeed because they implicitly model return distributions, similar to distributional RL (Bellemare et al., 2017). Indeed, much recent work applying flow-matching to value learning (Espinosa-Dice et al., 2025a; Dong et al., 2025) explicitly adopts distributional objectives. We test this hypothesis and find that it does not explain the observed gains: explicitly incorporating distributional updates often degrades performance relative to simple "expected-value" backups, which do not learn the return distribution. As such, flow-matching critics trained with standard TD consistently outperform strong distributional baselines. These results indicate that the advantage of flow matching does not come from modeling return distributions or distributional RL.

***So why do flow-matching critics work?*** In this paper, we show that their effectiveness stems from training a velocity network "wrapped" by an integration procedure (Figure 1), which is also used at inference time to predict Q-values. This form of *iterative computation* plays a dual role. First,

at inference time, the integration process enables the critic to recover from poor intermediate estimates and refine its prediction, yielding substantially more robust value estimates; we term this phenomenon **test-time recovery**. Second, beyond test-time recovery, training with the integration procedure induces more **plastic** features that allow better fitting subsequent TD targets. In particular, when trained with integration, the velocity network's representations need not change substantially to absorb successive TD targets, as these shifts can instead be accommodated through integration. This mitigates pathologies associated with loss of plasticity (Lyle et al., 2023) or representational capacity (Kumar et al., 2022). Put together, flow-matching critics outperform monolithic architectures with identical computation graphs that lack dense velocity supervision.

We support this argument empirically and theoretically. **Theoretically**, we formalize the notions of *test-time recovery* and the preservation of *plasticity* in simple settings, and show that flow-matching induces weight update dynamics that preserve and reweight previously learned features rather than overwriting them under non-stationary TD targets in linear settings. Such dynamics are absent in monolithic critics and their ensembles, and achieving similar behavior instead requires auxiliary objectives or explicit regularization that bias optimization and must be carefully tuned. **Empirically**, we show that flow-matching critics tolerate substantially higher noise, are more robust to interventions such as resetting or freezing network components, learn more isotropic features, and achieve much stronger performance, resulting in a $2\times$ performance gain and a $10\times$ improvement in sample efficiency in high update-to-data online RL with offline data. We further show that these gains arise from fitting velocities rather than directly regressing to TD targets.

## 2. Related Work

**Flow-matching in RL.** Flow-matching (Albergo & Vanden-Eijnden, 2023; Lipman et al., 2023) and diffusion are used to represent policies in RL (Ren et al., 2024; Celik et al., 2025; Ma et al., 2025; Lv et al., 2025; Wang et al.; Park et al., 2025b). More recently, flow-matching has been applied to value learning by parameterizing critics as velocity fields and training them via TD learning using both standard and distributional objectives (Agrawalla et al., 2025; Espinosa-Dice et al., 2025a; Dong et al., 2025). While prior work demonstrates strong empirical results, it remains unclear whether these gains stem from capacity, distributional modeling, or the learning dynamics induced by flow-matching (Agrawalla et al., 2025; Espinosa-Dice et al., 2025a; Dong et al., 2025). We provide evidence for the latter, showing that flow-matching yields more plastic features and enables robust value estimation via test-time recovery.

**Stability and plasticity in TD-learning.** Prior work has identified several pathologies in TD learning, including

value overestimation (Hasselt, 2010; Fujimoto et al., 2018), growth in parameter norms (Nikishin et al., 2022; Nauman et al., 2024a), and rapid loss of plasticity (Nikishin et al., 2023; Lyle et al., 2023) as training progresses. These issues are commonly attributed to bootstrapping and target non-stationarity, motivating a range of stabilization techniques, including architectural modifications such as layer normalization (Ba et al., 2016; Ball et al., 2023; Nauman et al., 2024b), weight or feature normalization (Kumar et al., 2022; Hussing et al., 2024; Lee et al., 2025), and alternative objectives such as cross-entropy style losses (Bellemare et al., 2017; Farebrother et al., 2024). Collectively, these approaches improve representational capacity and help maintain plasticity (Kumar et al., 2021; 2022; Lyle et al., 2024), and are widely used in modern RL algorithms (Kumar et al., 2023; Lee et al., 2025; Nauman et al., 2025; Palenicek et al., 2025). We show that flow-matching critics stabilize TD learning by addressing plasticity and robustness through the implicit bias of flow matching, without other regularization.

Finally, our results are related to work analyzing the learning dynamics of flow-matching models (Li & He, 2025; Bertrand et al., 2025). In contrast to these works, we study flow matching under the non-stationary TD-targets, and show that this difference leads to divergent conclusions and underscores the advantage of flow-matching.

## 3. Preliminaries, Notation, and Setup

Under usual notation of states and actions, we train a policy $\pi(a|s)$ that induces a distribution over the return random variable $Z^\pi(s, a) \triangleq \sum_{t=0}^{\infty} \gamma^t r(s_t, a_t)$. The expectation of $Z^\pi(s, a)$ is the Q-function of the policy: $Q^\pi(s, a) = \mathbb{E}[Z^\pi(s, a)]$. We focus on RL training with an offline dataset $\mathcal{D} = \{(s, a, r, s')\}$, in the fully offline RL (Levine et al., 2020) setting for most analysis. Value-based methods learn a network $Q_\theta(\mathbf{s}, \mathbf{a})$ by minimizing TD error.

**Flow-matching critics.** Flow-matching value functions depart from monolithic models that directly map $(\mathbf{s}, \mathbf{a})$ to a scalar. Instead, they represent values via a learned transformation of a random noise $\mathbf{z} \in \mathbb{R}$. Concretely, these methods parameterize a time-dependent velocity field $v_\theta(\mathbf{z}, t \mid \mathbf{s}, \mathbf{a})$ that defines an ODE over $\mathbf{z}$. Starting from an initial noise sample $\mathbf{z}_0 \sim p_0(\mathbf{z})$ at $t = 0$, numerical integration of this ODE ($\psi(t, \mathbf{z}|\mathbf{s}, \mathbf{a})$) produces a value sample at $t = 1$. Several recent works use this parameterization to learn *distributional* value functions. In particular, Espinosa-Dice et al. (2025a); Dong et al. (2025) train the velocity field so that integrating over the initial noise recovers the full return distribution $Z^\pi(\mathbf{s}, \mathbf{a})$. Given a transition $(\mathbf{s}, \mathbf{a}, r, \mathbf{s}') \sim \mathcal{D}$, a distributional TD target is constructed by pushing forward noise through the target flow at the next state:

$$\mathbf{z}' \sim p_0(\mathbf{z}), \; \tilde{Z}(\mathbf{s}, \mathbf{a}; \mathbf{z}') = r(\mathbf{s}, \mathbf{a}) + \gamma \, \psi_{\bar{\theta}}(1, \mathbf{z}'|\mathbf{s}', \mathbf{a}'), \; (1)$$

where $\psi_{\bar{\theta}}$ denotes the integrated target flow, $\mathbf{a}' \sim \pi(\cdot|\mathbf{s}')$.

Flow matching is then applied to align the velocity field with the transport from $\mathbf{z}$ (distinct from $\mathbf{z}'$) to samples from $\tilde{Z}(\mathbf{s}, \mathbf{a}; \mathbf{z}')$. A typical objective is given by $\mathcal{L}_{\text{dist}}(\theta) =$

$$\mathbb{E}_{\substack{(\mathbf{s},\mathbf{a},r,\mathbf{s}')\sim\mathcal{D}, \\ \mathbf{z},\mathbf{z}',\, t\sim\text{Unif}(0,1)}} \left[ \left\| v_\theta(\mathbf{z}(t), t \mid \mathbf{s}, \mathbf{a}) - \bar{s}_{\mathbf{z},\mathbf{z}'}(\mathbf{s}, \mathbf{a}) \right\|_2^2 \right], \quad (2)$$

where the target velocity is $\bar{s}_{\mathbf{z},\mathbf{z}'}(\mathbf{s}, \mathbf{a}) := \tilde{Z}(\mathbf{s}, \mathbf{a}; \mathbf{z}') - \mathbf{z}$, and the interpolant is $\mathbf{z}(t) = t \cdot \mathbf{z} + (1-t) \cdot \tilde{Z}(\mathbf{s}, \mathbf{a}; \mathbf{z}')$. This objective explicitly matches the *entire return distribution*.

`floq.` In contrast, Agrawalla et al. (2025) use flow matching to represent a Q-function while targeting only the *expected* return. Although `floq` integrates noise into a Q-value "sample" and therefore produces stochastic outputs during inference, its TD target collapses the next-state flow to a scalar expectation. Consequently, this approach does not learn or enforce a return distribution. Specifically, given $m$ i.i.d. noise samples $\{\mathbf{z}'_j\}_{j=1}^m$, define

$$y(\mathbf{s}, \mathbf{a}) := r(\mathbf{s}, \mathbf{a}) + \gamma \frac{1}{m} \sum_{j=1}^m \psi_{\bar{\theta}}(1, \mathbf{z}'_j \mid \mathbf{s}', \mathbf{a}'), \quad (3)$$

which estimates the expected-value TD target $r(\mathbf{s}, \mathbf{a}) + \gamma Q_{\bar{\theta}}(\mathbf{s}', \mathbf{a}')$. Flow-matching is then used to regress from initial noise $\mathbf{z} \sim \text{Unif}[l, u]$ to target $y(\mathbf{s}, \mathbf{a})$ via: $\mathcal{L}_{\text{floq}}(\theta) :=$

$$\mathbb{E}_{\substack{(\mathbf{s},\mathbf{a},r,\mathbf{s}')\sim\mathcal{D}, \\ \mathbf{z}\sim\text{Unif}[l,u], \\ t\sim\text{Unif}(0,1)}} \left[ \left\| v_\theta(\mathbf{z}(t), t \mid \mathbf{s}, \mathbf{a}) - \left( y(\mathbf{s}, \mathbf{a}) - \mathbf{z} \right) \right\|_2^2 \right]. \quad (4)$$

where the interpolant is $\mathbf{z}(t) = t \cdot \mathbf{z} + (1-t) \cdot y(\mathbf{s}, \mathbf{a})$. Thus, unlike the distributional objective above, `floq` does not learn or enforce a distributional Bellman equation; it uses flow matching purely as a *parameterization* of the expected Q-function. Equivalently, the target used to train the velocity field in Equation 4 can be viewed as the expectation of $s_{\mathbf{z},\mathbf{z}'}(\mathbf{s}, \mathbf{a})$ over the random variable $\mathbf{z}'$ alone.

Agrawalla et al. (2025) argue that `floq` is effective due to *iterative computation*, rather than distributional RL, a perspective also reflected in their training objective. Rather than fitting a Q-function in a single pass, they suggest that integration steps enable a gradual refinement of value estimates. While this explanation is appealing at a high level, it leaves open what iterative computation provides *formally*. In particular, if the benefits of flow matching arise solely from iterative computation at inference time, it is unclear why monolithic architectures fail to exhibit similar behavior.

*Our goal.* Our goal is to identify the mechanisms by which flow-matching Q-functions improve TD-learning. How does iterative integration interact with TD bootstrapping? Why does `floq` improve performance despite targeting only expected values? We argue that the answer lies not in distributional modeling, but in *test-time recovery* and improved *representational plasticity* enabled by flow-based critics, which we analyze in the remainder of the paper.

*Table 1.* ***Comparing expected-value (E) vs. distributional (D)*** `floq` on representative OGBench tasks. Each entry reports **E / D**. While both variants learn similar expected Q-values, **D** produces higher-variance estimates but does not higher performance than **E**.

| Env. | Success (%) | $Q_\theta(\mathbf{s},\mathbf{a})$ | $\text{Var}_{\mathbf{z}}(Q)$ |
|---|---|---|---|
| hmmaze-large | **52** ±8/30 ± 6 | -180 ±2/ − 170 ± 4 | 0.2 ±0.05/**4.5** ± 1 |
| antmaze-giant | **86** ±4/74 ± 4 | -190 ±3/ − 200 ± 2 | 0.1 ±0.02/**0.7** ± 0.1 |
| cube-double | 72 ±12/72 ± 10 | -130 ±8/ − 130 ± 4 | 1.1 ±0.2/**6.3** ± 1 |
| hmmaze-medium | 94 ±1/94 ± 2 | -170 ±2/ − 170 ± 2 | 0.3 ±0.02/**2.3** ± 0.3 |

## 4. Flow-Matching and Distributional RL

We now explicitly test whether distributional RL is necessary for strong performance of flow-matching critics. To do so, we use the TD-update from `floq` (Agrawalla et al., 2025) and modify it to use a distributional backup (Equation 2), also following Espinosa-Dice et al. (2025a). Importantly, we keep the velocity field architecture and hyperparameters identical across the two variants. We compare expected-value `floq` and its distributional counterpart on four representative OG-Bench tasks (Park et al., 2025a), along different axes: **(a)** the expected Q-value recovered on the offline dataset; **(b)** variance of the learned Q-value distribution; and **(c)** the performance of the learned policy.

**Results.** Observe in Table 1, that both expected and distributional variants recover nearly expected Q-values that are close to each other. However, the statistics of the learned Q-value distributions differ substantially. For instance, the standard deviation of the expected variant is significantly *lower* than that of the distributional variant. This behavior is consistent with the training objective. Since `floq` does not attempt to match the return distribution, and OG-Bench tasks are expected to exhibit high-variance, multimodal returns (Espinosa-Dice et al., 2025a; Dong et al., 2025), regressing to the expected-value target yields substantially lower-variance estimates. These results confirm that while both algorithms learn stochastic Q-function predictions, `floq` does not model the return distribution fully.

Despite this, in Table 1 we see that the distributional variant offers no benefits in performance and is often worse than expected-value `floq`. In addition, Agrawalla et al. (2025) also shows that `floq` outperforms strong distributional RL baselines such as C51 (Bellemare et al., 2017) and IQN (Dabney et al., 2018). Taken together, these findings demonstrate that flow-matching critics can perform extremely well even in the absence of distributional RL training, ruling it out as an explanation. These results also motivate our use of `floq` for the rest of the analysis.

> **Distributional RL is an insufficient explanation.**
>
> - Standard `floq` outperforms its distributional variant although it does not fit the return distribution, as it learns lower variance Q-value distributions.

# 5. Flow-Matching Enables Test-Time Recovery

So far, we have seen that the empirical gains of `floq` are not explained by distributional RL. Why, then, does flow matching work well? In this section, we develop a mechanistic understanding and mental model of how flow-matching Q-functions learn robust value estimates.

**Our main claim in this section** is that a flow-matching can *correct* imperfect intermediate estimates produced during the integration of the flow ODE. As more integration steps are performed, the final Q-value becomes less sensitive to errors made earlier in the integration process. We refer to this as *test-time recovery (TTR)*. Crucially, TTR does not arise from integration at test time alone, but from the *training procedure* used for flow-matching critics. Flow matching provides dense supervision of local velocity predictions along the entire integration trajectory, inducing a correction mechanism that is absent in monolithic critics. This perspective explains the two central design knobs of flow-matching critics: iterative computation at test time and dense supervision during training. Finally, we show that this training procedure also induces more *plastic* features, that put together with TTR explains the efficacy of this approach.

## 5.1. Flow-Matching Implements an Iterative Process

We begin by formalizing the notion of test-time recovery. Intuitively, test-time recovery describes the ability of a flow-matching Q-function to compensate for errors or inconsistencies introduced during inference, using a *single set of trained parameters*, by controlling the number of integration steps for computing Q-values. This mechanism is absent in monolithic critics that only query the network once.

---

**Definition 5.1** (*Test-Time Recovery*). Fix a state-action pair $(\mathbf{s}, \mathbf{a})$ and let $\psi_\theta(t, \mathbf{z}|\mathbf{s}, \mathbf{a})$ denote the flow interpolant obtained by integrating the velocity field $v_\theta$. Consider a $K$-step numerical integrator with step size $\eta = 1/K$ and discrete times $t_k = k/K$. Let $\{\psi^k\}_{k=0}^{K}$ be the unperturbed trajectory defined by

$$\psi^{k+1} = \psi^k + \eta\, v_\theta(\psi^k, t_k|\mathbf{s}, \mathbf{a}),$$

and let $\{\tilde{\psi}^k\}_{k=0}^{K}$ be a perturbed trajectory satisfying

$$\tilde{\psi}^{k+1} = \tilde{\psi}^k + \eta\big(v_\theta(\tilde{\psi}^k, t_k \mid \mathbf{s}, \mathbf{a}) + \xi_k\big),\ \tilde{\psi}^0 = \psi^0 = \mathbf{z},$$

where $\{\xi_k\}_{k=0}^{K-1}$ are *arbitrary* perturbations or errors incurred in velocity evaluations. We say that a trained flow-matching critic exhibits *test-time recovery* if the terminal error of the flow $\Delta_K(\mathbf{z}) := \tilde{\psi}^K - \psi^K$ satisfies: $\|\Delta_K(\mathbf{z})\| \leq \beta_K \sum_{k=0}^{K-1} \eta\, \|\xi_k\|$, for a stability factor $\beta_K < 1$ that decreases with $K$.

---

From a control-theoretic perspective, this definition of test-time recovery corresponds to *incremental stability* of the

inference-time dynamics induced by the learned velocity field, meaning that perturbations to intermediate states or velocity evaluations are progressively damped along the integration trajectory. We elaborate on this connection formally in Appendix D. Intuitively, TTR means that errors introduced at intermediate integration steps can be corrected by spending more integration steps. When $\beta_K$ decreases with $K$ (see Theorem D.1), increasing the number of integration steps improves robustness, explaining why flow-matching critics benefit from additional test-time compute.

*Why does TTR occur with flow-matching critics but is absent in monolithic networks?* Although Definition 5.1 characterizes an inference-time condition, flow-matching critics train a velocity network using TD-style supervision applied densely along the integration trajectory, across many interpolant inputs $\mathbf{z}$ and state-action pairs. As a result, the learned dynamics are explicitly shaped to correct local deviations at each integration step. Consequently, errors incurred early in the integration can be progressively attenuated, leading to more accurate Q-value estimates. This intuition is supported by Definition 5.1, which shows that even when the per-step TD errors of a flow-matching critic are comparable to those of a monolithic critic, the iterative integration mechanism can yield a more robust final value estimate. In contrast, monolithic critics, including ResNet-style architectures with similar inference-time computation graphs, are trained with terminal supervision only and lack any mechanism to control intermediate representations or suppress error propagation. As a result, errors introduced at intermediate layers tend to accumulate. Finally, we show that this same dense supervision also improves the learned features of the velocity network when chasing non-stationary TD targets, allowing the model to adapt by preserving previous features, which further strengthens the TTR phenomenon.

## 5.2. Analysis: Test-Time Recovery (TTR) in Practice

We now evaluate the ability of flow-matching critics to perform test-time recovery. To do so, we introduce controlled perturbations into the integration process and measure how sensitive the Q-value estimates are to these interventions. For comparison, we also construct analogous perturbations for monolithic critics and evaluate their behavior.

*Experiment:* **Injecting staleness into earlier integration steps of the flow critic.** In this experiment, we split the integration procedure into two phases. After training a flow-matching critic for $T = 250,000$ gradient steps, we evaluate a modified procedure in which the first $\kappa\%$ of the integration steps ($\kappa \in \{0, 25, 50, 75, 100\}$) are intentionally performed using a stale snapshot of the velocity field taken from the checkpoint at time $T$, while the remaining steps are completed using the current network parameters. We compare this procedure to a baseline which runs all integration steps with current parameters ($\kappa = 0$). If flow-matching exhibits

*Table 2.* ***Effect of injecting staleness into early integration steps*** of the flow-matching critic. Entries are shaded by performance degradation relative to the best value within each environment as $\kappa = \{0, 25, 50, 75, 100\}\%$ increases from left to right. Note that a flow-matching critic *can* still succeed in many cases when the first 25% or even 50% of the integration is done with a stale velocity field. On 2/3 environments, using a stale velocity field improves performance slightly (best performance in each row is in **bold**). We also report standard error across seeds (in brackets) with the performance.

| Env. (default task) | Success as stale fraction $\kappa$ increases |
|---|---|
| antsoccer-arena | $45(4) \rightarrow \mathbf{50}(4) \rightarrow 47(3) \rightarrow 46(4) \rightarrow 35(4)$ |
| hmmaze-medium | $\mathbf{98}(2) \rightarrow 43(4) \rightarrow 63(20) \rightarrow 62(15) \rightarrow 39(10)$ |
| cube-double | $72(15) \rightarrow \mathbf{84}(10) \rightarrow 82(15) \rightarrow 58(10) \rightarrow 64(5)$ |

test-time recovery, then later integration steps should be able to correct *at least some* errors arising from staleness of the early steps of the integration trajectory, even though the parameters of the velocity field cannot be updated.

**Results.** Table 2 provides empirical evidence of test-time recovery. On 2 of 3 environments, using stale velocity parameters for the first 25% or 50% of the integration steps in the critic yields policies with higher success rates than using no stale velocities at all, indicating that flow-matching critics can recover from errors introduced early during integration. This behavior is not universal, as on the `humanoidmaze-medium` task any amount of staleness leads to substantial performance degradation.

In contrast, analogous interventions applied to monolithic feed-forward critics consistently result in pronounced performance drops, as shown in Figure 4. We observe similar degradation for ResNet-style critics with residual connections (Figure 5), suggesting that a computation graph resembling an integration process is insufficient to induce TTR. Instead, recovery requires training with a flow-matching loss itself. Finally, as we discuss in the next section, flow matching also induces more robust feature representations, explaining why flow-matching critics remain resilient not only to stale velocities but also to stale internal features.

*Experiment:* **Robustness to noisy TD supervision.** In the previous experiment, we perturbed the integration procedure itself. Here, we examine a complementary probe in which we corrupt the *training-time supervision*, i.e., TD targets that supervise the velocity field. Of course, this corruption would degrade the performance of any approach, but if flow-matching does indeed exhibit TTR, it should exhibit a more graceful degradation as the magnitude of corruption increases. We add noise to the velocity network targets as per the procedure in Appendix C.1. This noise directly affects the local supervision signal at each integration step. We also train a monolithic Q-network baseline with target noise and present its results below.

**Results.** Figure 2 shows that performance degradation is more graceful for the flow-matching critic, which consis-

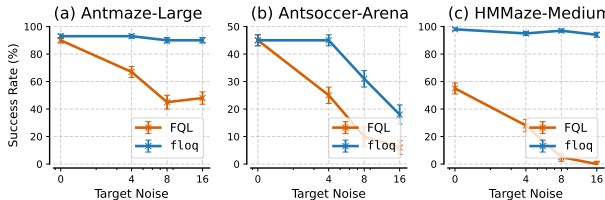

*Figure 2.* ***Performance of flow-matching (`floq`) and monolithic (FQL) critics when trained with target noise***. Observe that flow-matching critics are much more robust to noise in TD-targets, while performance of FQL (monolithic critics) degrades substantially faster, even when they start at a similar point (antmaze/antsoccer).

tently maintains higher performance (and even no degradation) than the monolithic approach as noise magnitude increases. These results suggest that robustness to noisy supervision can arise in flow-matching critics. Overall, we see that the learned integration dynamics of flow-matching critic remains more stable, allowing later integration steps to partially attenuate the effect of noisy supervision.

> **Test-time recovery in flow-matching critics.**
>
> - Flow-matching critics exhibit *test-time recovery*: perturbations to integration or TD targets dampen as more integration steps are performed.
> - Supervising velocities is crucial for effective TTR; monolithic architectures with similar computation graphs (e.g., ResNets) do not exhibit TTR.

## 6. Flow-Matching Learns Plastic Features

The experiments in the previous section show that flow-matching critics exhibit test-time recovery, and that this behavior arises from supervising velocities along the integration trajectory rather than from the computation graph alone. We now ask whether this same structure also shapes training dynamics. In this section, we show that training a velocity field jointly with the integration procedure leads to more *plastic* representations, which are crucial for mitigating the pathologies induced by non-stationary TD targets.

**Intuition.** Non-stationary TD targets computed on unseen actions can lead to pathologies such as feature rank collapse, exploding feature norms and dead neurons, as critics must repeatedly overwrite their features to track moving targets, eventually exhausting representational capacity (Kumar et al., 2021; Lyle et al., 2022; Nikishin et al., 2022). We hypothesize that flow-matching critics alleviate these issues due to their structure. Rather than requiring the velocity network to directly update its features to chase each new TD target, flow matching allows the integration process to absorb large changes in value estimates, reducing the need for major changes to the network's internal representations. Thus, the learned features retain greater plasticity over the course of training. ***Viewed through this lens, integration plays a dual role:*** *at test time*, it enables recovery and ro-

bust Q-value estimates; *at training time*, it acts as a buffer between changing TD targets and learned representations, promoting more stable feature learning. We now present evidence to support this intuition.

### 6.1. Theoretical Analysis in the Linear Setting

To understand why flow-matching critics preserve plasticity under non-stationary TD targets, we analyze a simple yet informative linear setting. We consider TD-learning with a *linear* flow-matching critic and compare it to a corresponding "deep" linear ResNet (Arora et al., 2018). Although both models represent linear predictors with comparable computation graphs, their learning dynamics differ.

**Linear model setup.** We consider learning linear predictors on inputs $\mathbf{x} \in \mathbb{R}^d$, when training against non-stationary targets. We denote the non-stationary targets as $y(m)$, where $m$ denotes the training step. A ***monolithic*** critic is given by $f_{\mathrm{mono}}(\mathbf{x}; m) \triangleq w(m)^\top \mathbf{x}$, where $w(m) \in \mathbb{R}^d$ is the effective weight and $\mathbf{x}$ compactly represents the concatenated input $[\mathbf{s}, \mathbf{a}]$. The effective weight vector $w(m)$ is given by a product of multiple linear layers, i.e., $w(m) = \prod_{t=1}^{T} u_t(m)$. A deep linear ResNet (He et al., 2015) is given by: $w(m) = \prod_t (I + u_t(m))$. Any monolithic critic directly trains the parameters in $w(m)$ against the non-stationary $y(m)$ by minimizing squared error via gradient descent.

A ***flow-matching*** critic with a comparable linear architecture can be characterized explicitly by unrolling the integration process. Since the interpolant at each step depends on the output of the previous step, the resulting mapping is recursive in the input $\mathbf{x}$. Consider a linear velocity network at each integration step. Let $\{u_t(m)\}_{t=1}^{T-1}$ denote the linear velocity slices and let $\{\alpha_t\}_{t=1}^{T-1}$ denote the integration step sizes. Then the output velocity field is: $v_\theta(\mathbf{z}, t | \mathbf{x}) = v_t(m) \cdot \mathbf{z} + u_t(m)^\top \mathbf{x}$. We refer to $u_t(m)$ as the weights parameterizing the $t$-th "slice" and $v_t(m)$ denotes the gain parameters that multiply the interpolant.

Unrolling the $T$-step Euler integration yields the following *expected* output of the integration process. Here, expectation is taken over the initial noise $\mathbf{z}$. If the noise mean is non-zero, an additional bias term independent of $\mathbf{x}$ appears, which we remove for clarity as it does not change any of our conclusions.

$$f_{\mathrm{FM}}(\mathbf{x}; m) = \sum_{t=1}^{T-1} \beta_t(m) \cdot \underbrace{u_t(m)^\top \mathbf{x}}_{\text{Output from } t\text{-th linear "slice"}}, \quad (5)$$

where the integration-dependent amplification coefficient is $\beta_t(m) \triangleq \alpha_t \prod_{k=t+1}^{T-1} (1 + \alpha_k v_k(m))$. Here $\{v_k(m)\}_{k=1}^{T-1}$ denote scalar "gain" parameters induced by recursive integration. The gains $v_k(m)$ quantify how the contribution of the $t$-th velocity computation propagates through subsequent integration steps to influence the final Q-value prediction.

---

**Theorem 6.1** (**Flow critics can learn by reweighting existing features; monolithic critics must modify features.**). *Consider training monolithic and flow-matching models by minimizing squared error against a non-stationary target $y(m)$. Fix an interval of training steps $m \in [m_0, m_1]$ and suppose*

$$\dot{u}_t(m) = 0 \quad \text{for all } t \text{ and } m \in [m_0, m_1],$$

*i.e., feature directions are frozen during this training interval. Then the following hold:*

1. *(**Monolithic**). If $\frac{d}{dm} f_{\mathrm{mono}}(\cdot; m) \neq 0$ on $m \in [m_0, m_1]$, then necessarily $\exists u_t$ s.t. $\dot{u}_t(m) \neq 0$.*

2. *(**Flow-matching**). The Euler flow-matching predictor satisfies*

$$\frac{d}{dm} f_{\mathrm{FM}}(\mathbf{x}; m) = \Big( \sum_{t=1}^{T-1} \dot{\beta}_t(m) \, u_t(m) \Big)^\top \mathbf{x},$$

*where*

$$\dot{\beta}_t(m) = \beta_t(m) \sum_{k=t+1}^{T-1} \frac{\alpha_k \, \dot{v}_k(m)}{1 + \alpha_k v_k(m)}.$$

*Thus, even when feature directions $\{u_t(m)\}$ remain **all** fixed, the predictor can adapt through changes in the gain parameters $\{v_t(m)\}$, i.e., $\dot{v}_k(m) \neq 0 \Rightarrow \dot{\beta}_t(m) \neq 0, \ \forall t < k.$*

---

Because each integration step multiplicatively rescales the interpolant via a factor depending on $v_k(m)$, earlier slices can be amplified or attenuated depending on downstream gain dynamics, leading to a structured reweighting of feature directions. Thus, although $f_{\mathrm{FM}}$ is linear in $\mathbf{x}$, its effective weight vector

$$w_{\mathrm{FM}}(m) = \sum_{t=1}^{T-1} \beta_t(m) \, u_t(m) \quad (6)$$

is not freely parameterized. Instead, it decomposes into velocity slices whose coefficients are coupled through the integration procedure. We argue in Appendix E that monolithic networks can adapt only by updating $w(m)$ (and hence, each of $u_t(m)$ since there is one supervision on the entire weight $w(m)$), whereas flow-matching critics can adapt by reweighting existing features $\{u_t(m)\}$ through changes in the gain dynamics $\{v_t(m)\}$ due to dense supervision. Our theoretical result for this setting is given in Theorem 6.1. For completeness, we extend Theorem 6.1 to monolitihic ensembles in Appendix E.8 and moving features in Appendix F, and identify similar gains over monolithic critics.

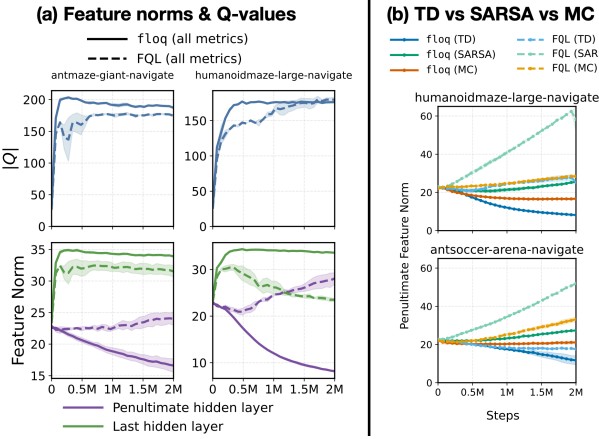

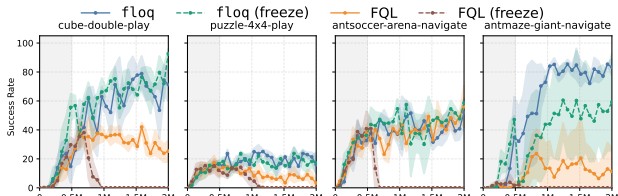

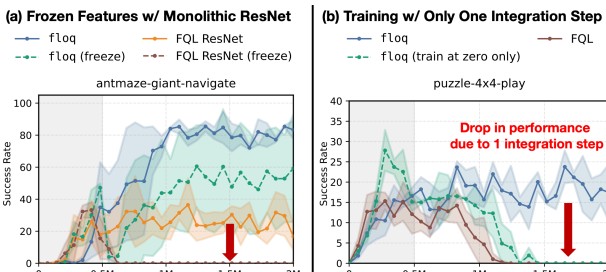

*Figure 3.* **Feature norms.** **(a)** Learned feature norms and average Q-values for monolithic critics (FQL) and flow-matching critics (`floq`) in the penultimate and last hidden layers. While the last hidden layer adapts to the scale of Q-values for both methods, the penultimate hidden layer in `floq` exhibits a much more rapid decrease in feature norms compared to FQL. This indicates that `floq` learns more flexible and adaptive features in the penultimate hidden layer that are largely decoupled from the magnitude of Q-values. **(b)** Penultimate hidden layer feature norms for `floq` trained with TD, SARSA, and MC targets. `floq` with TD shows the fastest decrease in feature norms, whereas SARSA and MC trends resemble those of the monolithic FQL critic. This suggests that flow-matching critics, particularly under TD learning, develop more robust representations under non-stationary targets.

### 6.2. Empirical Evidence of Plastic Feature Learning

To empirically test this intuition, we examine whether flow-matching critics learn more plastic and qualitatively different feature representations than monolithic networks. In particular, we ask whether the interaction between flow matching and TD-learning leads to differences in learned representations relative to monolithic critics, and whether such differences are specific to TD-learning. We run several experiments that we describe in the paragraphs below.

*Experiment:* **Measuring properties of learned features.** We measure the $\ell_2$-norm of *post-layernorm* features learned by the velocity network (for flow critics) and the critic network (for monolithic critics) for three learning algorithms: **(a)** TD-learning, **(b)** SARSA (using the dataset action for the TD backup), and **(c)** regression to pre-computed Monte Carlo (MC) returns. As shown in Figure 3 (ref. Fig 8 for more tasks), across several tasks, flow-matching critics trained with TD-learning learn much lower-norm penultimate hidden-layer features than monolithic critic networks, despite the absence of any explicit regularization, which has in fact appeared as a desirable property (Kumar et al., 2023; Hussing et al., 2024). This also indicates that a significant burden of modeling the Q-value scale is deferred to the final layer and the integration, rather than encoded through the network. As such, the model is less likely to learn spurious features to explain the changes in magnitude of TD targets.

Notably, this effect is absent when training with SARSA

*Figure 4.* **Measuring feature plasticity** on four OG-bench tasks, by freezing all layers except the final two at $T = 0.5M$ steps (gray shaded region denotes the pre-freeze phase). Solid curves correspond to the default (fully trained) runs, while dashed curves show performance after freezing the penultimate hidden features. Across all environments, *FQL with a monolithic critic exhibits a sharp performance collapse* once features are frozen (brown vs orange), indicating in inability to represent future Q-functions. In contrast, flow-matching critics remain stable and continue to improve after freezing, demonstrating substantially greater plasticity.

or Monte-Carlo (MC) regression (Figure 3) (ref. Fig 9 for more tasks), where both flow-matching and monolithic critics exhibit similar feature statistics. This suggests that the representational differences arise specifically from the interaction between flow matching and TD-learning, and not from architectural differences alone. In particular, TD-learning relies on bootstrapping from policy actions that may be out-of-distribution for the offline dataset, introducing a higher degree of non-stationarity in the targets than SARSA.

*Figure 5.* **(a)** *Frozen features with a monolithic ResNet critic.* Although a monolithic ResNet admits a computation graph similar to a flow-matching critic, freezing its features leads to a collapse in performance during subsequent offline RL training. **(b)** *Frozen features with a single integration step.* With only one integration step, a flow-matching critic is more stable than a monolithic network, but less stable than full flow matching with multiple integration steps (performance drop indicated by the red arrow), highlighting the essential role of integration in preserving feature plasticity.

*Experiment: Measuring feature plasticity by freezing features.* To assess whether these representational differences are consequential, we probe feature plasticity by freezing the early layers of the critic at an intermediate point during offline training and continuing TD-learning on the offline dataset. If the learned features are sufficiently expressive to support future TD targets, the impact of freezing should diminish with further training. As shown in Figure 4, freezing features causes monolithic critics to collapse to near-zero performance across almost all environments, with little to no recovery. In contrast, flow-matching critics recover to performance comparable to the unfrozen baseline

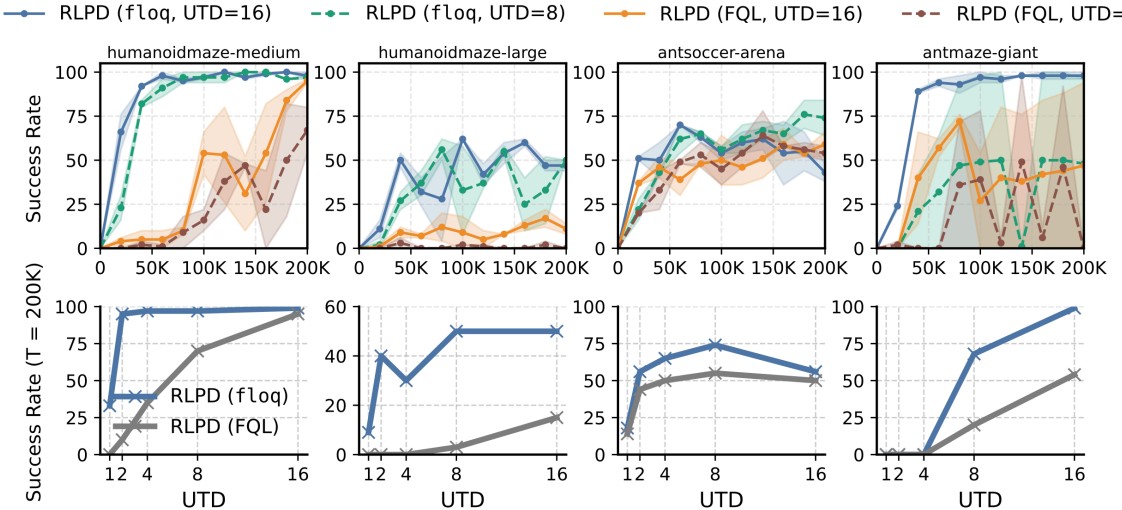

*Figure 6.* **High-UTD online RL with offline data.** We compare flow-matching critics (`floq`) to monolithic critics (FQL) under the RLPD framework, using a 50:50 mixture of offline data and online rollouts. Flow-matching critics consistently achieve higher performance across all UTD values (bottom row) and substantially improve sample efficiency, with gains of up to 5-10× in some environments.

on all but one environment, where performance remains competitive. This indicates that features learned by flow-matching critics remain useful for representing *future value functions* that will be encountered, even when these features are no longer updated. This supports the notion that flow-matching learns features that help model the value improvement path (Dabney et al., 2020). We further confirm this behavior by applying the same intervention to a monolithic critic with a ResNet-style architecture (Figure 5), which also collapses once intermediate layers are frozen.

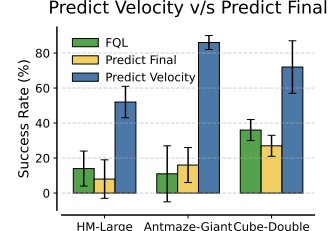

To isolate the role of integration, we additionally evaluate a flow-matching critic trained with only a single integration step ("train at zero only"), where no integration is performed at inference time. While this variant outperforms a monolithic critic (FQL) after the point of intervention (Figure 5; Figure 10 for more tasks), it is substantially less stable than full flow matching, highlighting the crucial role of integration in enabling plastic representations. Finally, we note that unlike in online RL, where learning can recover from frozen/reset features by collecting new data online (Nikishin et al., 2022), such mechanisms are unavailable in offline RL.

*Figure 7.* **Impact of predicting velocities versus final TD-targets** at each integration step. Predicting the final TD-target substantially degrades the performance of a flow-matching critic, yielding performance comparable to monolithic critics. This underscores the importance of dense velocity supervision in flow-matching critics.

*Experiment:* **Intermediate velocity supervision is crucial.**

> **Flow-matching critics learn plastic features.**
>
> - Flow-matching critics learn more plastic representations than monolithic critics: freezing features severely degrades monolithic critics but has little effect on flow-matching critics.
> - Supervising *velocities* is essential for these benefits; directly supervising absolute TD targets collapses flow matching to monolithic behavior, eliminating test-time recovery and plasticity.

Finally, we test whether the specific choice of supervising the *velocity field*, rather than absolute TD targets, is crucial for obtaining benefits of flow matching discussed so far. Motivated by recent work (Li & He, 2025), we implement a variant of flow matching in which each integration step is supervised to directly predict the TD target value itself, rather than the velocity. As shown in Figure 7, although this variant still uses integration and receives dense supervision at every integration step, it fails to retain the benefits of flow matching degrading to a performance comparable to that of a monolithic critic. Empirically, we find that the network learns to ignore the interpolant and instead fit the target values independently at each step, effectively collapsing to an ensemble of monolithic critics. As a result, performance degrades and the gains from iterative computation disappear. This behavior highlights that training a velocity field together with the integration "wrapper" is critical for attaining better performance.

## 7. High-UTD Online RL with Prior Data

The analysis above shows that the gains of flow matching arise from training the velocity field jointly with the integration procedure. Given non-stationary TD targets,

flow-matching enables test-time recovery and learns more plastic features. Here, we examine whether these properties improve performance in high update-to-data (UTD) regimes. While aggressive data reuse is known to improve efficiency, it often degrades feature plasticity and destabilizes learning (Chen et al., 2021; D'Oro et al., 2022). If a flow-matching critic indeed maintains adaptability under non-stationary TD targets, its performance should scale faster when increasing UTD. To test this, we incorporate flow-matching critics into the RLPD framework (Ball et al., 2023), enabling more aggressive data reuse. As shown in Figure 6, high-UTD training with `floq`, a flow-matching critic, achieves substantially stronger performance at large UTD ratios, with roughly $2\times$ higher final return and a 5-$10\times$ improvement in sample efficiency compared to an RLPD baseline on top of FQL, using monolithic critics. We also find that flow-matching critics are more stable and do not destabilize, even at the highest UTD value we tested. These results show that the plastic features learned via flow-matching enable efficient learning in high UTD settings.

**Discussion and Perspective on Future Work.** Please see Appendix G for a discussion of the main findings in the paper and the associated future work.

## Impact Statement

This work contributes to a better understanding of flow-matching methods in reinforcement learning (RL). We do not foresee any direct societal impacts specific to this work beyond those generally associated with advances in reinforcement learning research.

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

# Appendices

## A. Additional Results for Flow-Matching Critics

**Post-layerNorm feature Norms for flow-matching critics (`floq`) vs monolithic critics (FQL).** We show in Figure 8 that while the last hidden layer adapts to the $Q-$ scale for both `floq` and monolithic critics, the penultimate hidden layer in `floq` exhibits a rapid decrease in feature norms compared to FQL. This indicates that `floq` learns more adaptive representations in the penultimate hidden layer that are largely decoupled from the Q-value scale.

**Penultimate hidden layer feature norms comparison for `floq` (TD), `floq` (SARSA) and `floq` (MC).** We show in Figure 9 that `floq` trained with TD shows the fastest decrease in penultimate hidden layer feature norms, whereas `floq` (SARSA) and `floq` (MC) trends resmeble those of the monolithic FQL critics. This suggests that flow-matching critics develop better representations under non-stationary TD targets.

**Frozen features with a single integration step.** We show in Figure 10 that flow-matching critics with one integration step are more stable than monolithic critics, but fall short in learning plastic features than full flow matching critics. This highlights that integration plays a crucial role in preserving feature plasticity.

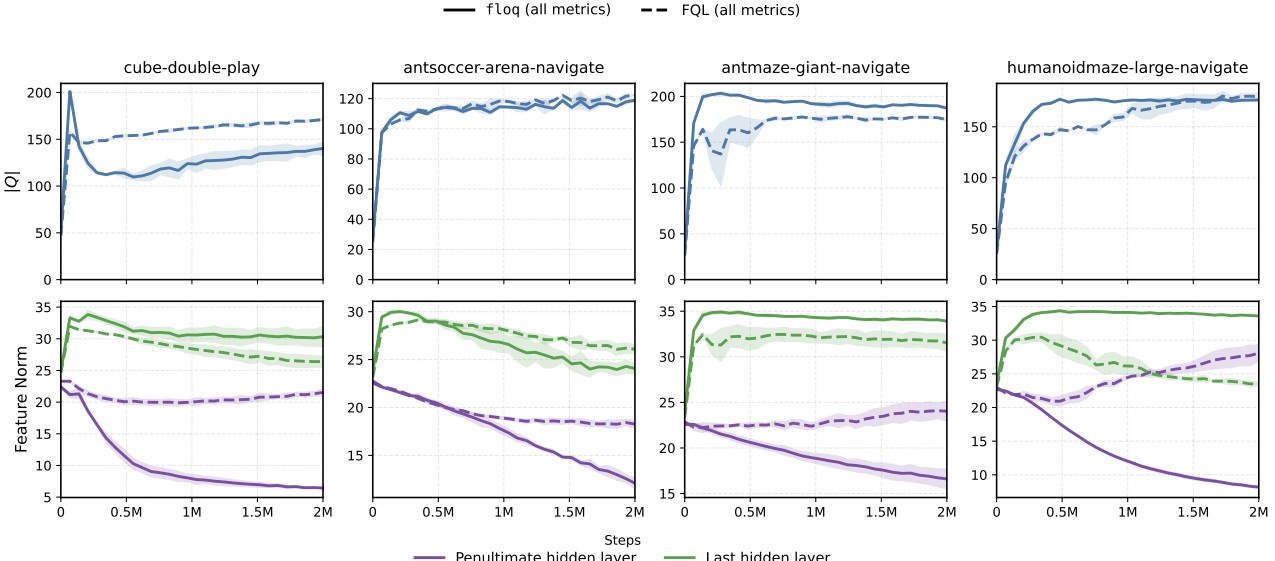

*Figure 8.* While the last hidden layer adapts to the Q-value scale for both `floq` and monolithic critics, the penultimate hidden layer layer in `floq` exhibits a qualitatively distinct, rapid decrease in feature norms. This indicates that `floq` learns more adaptive representations in the penultimate hidden layer that are largely decoupled from the Q-value scale.

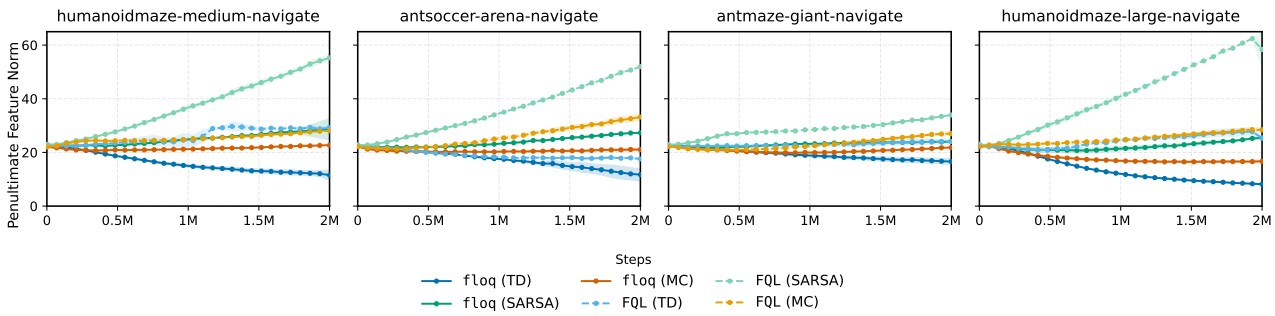

*Figure 9.* Flow-matching critics trained with TD-learning (`floq`) shows the fastest decrease in penultimate hidden layer feature norms, whereas `floq` (SARSA) and `floq` (MC) trends resmeble those of the monolithic FQL critics. Thus flow-matching critics develop particularly better representations under non-stationary TD targets.

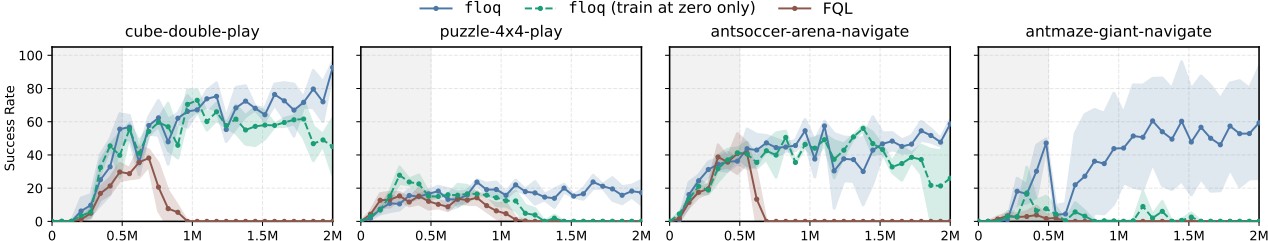

*Figure 10.* Flow-matching critics with one integration step are more stable than monolithic critics but less stable than full flow matching. This emphasizes that test-time recovery in the form of integration plays a crucial role in preserving feature plasticity.

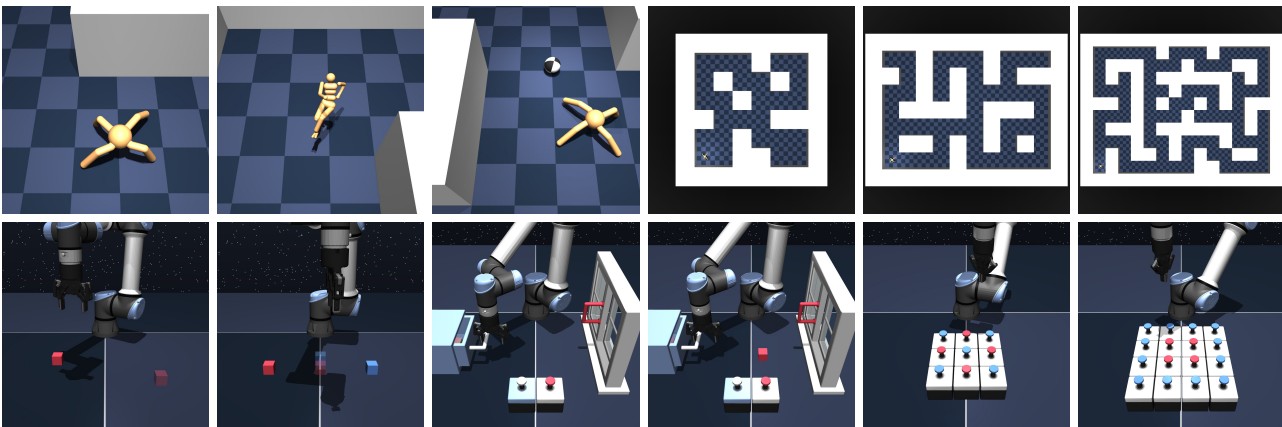

*Figure 11.* **OGBench (Park et al., 2025a) domains**. These tasks include high-dimensional state and action spaces, sparse rewards, stochasticity, as well as hierarchical structure.

## B. Benchmarks

Following evaluation protocols from recent work in offline RL (Park et al., 2025b; Wagenmaker et al., 2025; Espinosa-Dice et al., 2025b;a; Dong et al., 2025), we use the **OGBench** task suite (Park et al., 2025a) as our main evaluation benchmark (see Figure 11). OGBench provides a number of diverse, challenging tasks across robotic locomotion and manipulation, where these tasks are generally more challenging than standard D4RL tasks (Fu et al., 2020), which have been saturated as of 2024 (Tarasov et al., 2023; Rafailov et al., 2024; Park et al., 2024). While OGBench was originally designed for benchmarking offline goal-conditioned RL, we use its reward-based single-task variants ("`-singletask`" from Park et al. (2025b)).

We used the `default` task in each OGBench environment for our analysis and RLPD experiments, following the protocol in recent works (Park et al., 2025b; Wagenmaker et al., 2025; Espinosa-Dice et al., 2025b;a; Dong et al., 2025).

## C. Experimental Details and Hyperparameters

### C.1. Offline RL Analysis Experiments (Section 4, Section 5, Section 6).

**Experimental details for robustness to noisy TD supervision (Figure 2, Section 5.2).** We added $\text{Unif}[-\kappa, \kappa]$ (for $\kappa \in \{0, 4, 8, 16\}$) noise to the $Q$-network targets for FQL (monolithic critics) and to the velocity-network targets for `floq` (flow-matching critics).

**Hyperparameters.** We used the hyper-parameters from (Agrawalla et al., 2025) for both `floq` (flow-matching critics) and FQL (monolithic critics) for all offline RL analysis experiments.

### C.2. RLPD Experiments (Section 7).

We tuned the BC-regularization coefficient ($\alpha$) in the range $[10, 100]$ (step size 10) for both `floq` and FQL. All other hyper-parameters were kept to the same values from (Agrawalla et al., 2025).

## D. Theoretical Result for Test-Time Recovery (Section 5.1).

**Theorem D.1.** *Fix a state-action pair* $(\mathbf{s}, \mathbf{a})$ *and let* $\psi_\theta(t, \mathbf{z}|\mathbf{s}, \mathbf{a})$ *denote the flow interpolant obtained by integrating the velocity field* $v_\theta$. *Assume that the velocity field* $v_\theta$ *satisfies*

$$\frac{\partial v_\theta(x, t; s, a)}{\partial x} \leq -\frac{c_1}{1 - t}.$$

*for some constant* $0 < c_1 < 1$ *and all* $t \in (0, 1 - \frac{1}{K})$.

*Consider a* $K$-*step numerical integrator with step size* $\eta = 1/K$ *and discrete times* $t_k = k/K$. *Let* $\{\psi^k\}_{k=0}^K$ *be the unperturbed trajectory defined by*

$$\psi^{k+1} = \psi^k + \eta\, v_\theta(\psi^k, t_k | \mathbf{s}, \mathbf{a}),$$

*and let* $\{\tilde{\psi}^k\}_{k=0}^K$ *be a perturbed trajectory satisfying*

$$\tilde{\psi}^{k+1} = \tilde{\psi}^k + \eta\big(v_\theta(\tilde{\psi}^k, t_k \mid \mathbf{s}, \mathbf{a}) + \xi_k\big), \quad \tilde{\psi}^0 = \psi^0 = \mathbf{z},$$

*where* $\{\xi_k\}_{k=0}^{K-1}$ *are arbitrary perturbations (or errors) in velocity evaluations. Assume further that each* $\|\xi_k\| \leq \varepsilon$ *for some* $\varepsilon > 0$.

*Then the terminal error* $\Delta_K(\mathbf{z}) := \tilde{\psi}^K - \psi^K$ *satisfies*

$$\|\Delta_K(\mathbf{z})\| \leq \beta_K \varepsilon$$

*where*

$$\beta_K = c_2 K^{-c_1}$$

*for some constant* $c_2 > 0$.

**Note.** We argue here that assumption on $v_\theta$ is reasonable for flow-matching networks (by choosing any appropriate $c_1 < 1$).

This is because the learned trajectories are straight to the first order (the curvature ,while being important to distinguish actions, appears at the scale of advantages; which are typically much smaller than the $Q$-value - see Figure 16 in (Agrawalla et al., 2025) for an example). Moreover, the variance in flow outputs is much smaller than the range of initial noise for `floq` style expected-value training (see Table 1).

This implies that to the first order, we have

$$v_\theta(x, t; s, a) \approx \frac{Q(s, a) - x}{1. - t} \implies \frac{\partial v_\theta(x, t; s, a)}{\partial x} \approx -\frac{1}{1. - t} \leq -\frac{c_1}{1. - t}$$

for an appropriate constant $c_1 < 1$.

*Proof.* We analyze the evolution of the error between the unperturbed trajectory $\psi^k$ and the perturbed trajectory $\tilde{\psi}^k$.

**1. Error Dynamics** Let $\Delta_k = \tilde{\psi}^k - \psi^k$. By subtracting the update equations for the two trajectories, we obtain:

$$\begin{aligned}
\Delta_{k+1} &= \tilde{\psi}^{k+1} - \psi^{k+1} \\
&= (\tilde{\psi}^k - \psi^k) + \eta\left[v_\theta(\tilde{\psi}^k, t_k \mid \mathbf{s}, \mathbf{a}) - v_\theta(\psi^k, t_k \mid \mathbf{s}, \mathbf{a})\right] + \eta\xi_k \\
&= \Delta_k + \eta\left[v_\theta(\tilde{\psi}^k, t_k \mid \mathbf{s}, \mathbf{a}) - v_\theta(\psi^k, t_k \mid \mathbf{s}, \mathbf{a})\right] + \eta\xi_k
\end{aligned}$$

**2. Applying the Mean Value Theorem** By the Mean Value Theorem (MVT) for vector-valued functions, the difference in the velocity fields can be expressed as:

$$v_\theta(\tilde{\psi}^k, t_k) - v_\theta(\psi^k, t_k) = \left(\int_0^1 \nabla_x v_\theta(\psi^k + \tau\Delta_k, t_k)\, d\tau\right)\Delta_k$$

Applying the assumption $\frac{\partial v_\theta}{\partial x} \leq -\frac{c_1}{1-t}$, and substituting $t_k = k/K$ and $\eta = 1/K$:

$$\Delta_{k+1} \leq \left(1 - \frac{\eta c_1}{1 - t_k}\right)\Delta_k + \eta \xi_k$$

$$= \left(1 - \frac{(1/K)c_1}{1 - (k/K)}\right)\Delta_k + \eta \xi_k$$

$$= \left(1 - \frac{c_1}{K - k}\right)\Delta_k + \eta \xi_k$$

**3. Unrolling the Recurrence**  Since $\tilde{\psi}^0 = \psi^0$, we have $\Delta_0 = 0$. Unrolling the recurrence from $k = 0$ to $K - 1$:

$$\|\Delta_K\| \leq \sum_{k=0}^{K-1}\left(\prod_{j=k+1}^{K-1}\left(1 - \frac{c_1}{K - j}\right)\right)\eta\|\xi_k\|$$

$$\leq \frac{\varepsilon}{K}\left[[(1 - c_1)] + [(1 - c_1)(1 - \frac{c_1}{2})] + [(1 - c_1)(1 - \frac{c_1}{2})(1 - \frac{c_1}{3})] + \cdots + [(1 - c_1)(1 - \frac{c_1}{2})(1 - \frac{c_1}{3})\ldots(1 - \frac{c_1}{K})]\right]$$

$$\leq \frac{\varepsilon(c_2 . K^{1-c_1})}{K}$$

$$\leq (c_2 K^{-c_1})\varepsilon$$

for a constant $c_2 > 0$, as desired. $\qquad\square$

# E. Comparing Flow-Matching and Monolithic Networks in the Linear Setting (Section 6.1)

Fix $T \geq 3$ and step size $h := \frac{1}{T-1}$. Consider the linear Euler recursion

$$s_{i+1}(x, z; t) = (1 + hv_i(t))s_i(x, z; t) + h\, u_i(t)^\top x, \qquad i = 1, \ldots, T - 1, \tag{7}$$

$$s_1(x, z; t) = z, \qquad \hat{y}(x, z; t) := s_T(x, z; t), \tag{8}$$

with parameters $u_i(t) \in \mathbb{R}^d$, $v_i(t) \in \mathbb{R}$.

**Gain parameters.**  In the Euler recursion

$$s_{i+1} = (1 + hv_i)s_i + hu_i^\top x,$$

the scalar parameter $v_i$ is a *gain*: it multiplicatively scales the incoming state before propagation. Positive gain amplifies the accumulated state, while negative gain attenuates it. Because the end-to-end contribution of an injected feature is

$$\beta_m = h\prod_{j>m}(1 + hv_j),$$

downstream gains determine how much signal injected at earlier slices survives to the output. Learning the gains therefore corresponds to learning how strongly the model should trust previously accumulated information.

## E.1. Unrolled Predictor

**Lemma E.1** (Unrolled form). *Define*

$$P_{>m}(t) := \prod_{j=m+1}^{T-1}(1 + hv_j(t)), \qquad \beta_m(t) := h\, P_{>m}(t).$$

*Then*

$$\hat{y}(x, z; t) = \left(\prod_{j=1}^{T-1}(1 + hv_j(t))\right)z + \sum_{m=1}^{T-1}\beta_m(t)\, u_m(t)^\top x. \tag{9}$$

*Consequently, if $\mathbb{E}[z] = 0$,*

$$f(x;t) := \mathbb{E}_z[\hat{y}(x,z;t)] = \sum_{m=1}^{T-1} \beta_m(t)\, u_m(t)^\top x. \tag{10}$$

## E.2. Flow-Matching Gradient Flow

Let $Y(t)$ be a time-dependent target and $Z$ independent noise with $\mathbb{E}[Z] = 0$, $\text{Var}(Z) = \sigma_z^2$. Define

$$\alpha_i := \frac{T-i}{T-1}, \qquad S_i(t) := \alpha_i Z + (1 - \alpha_i)Y(t).$$

Each slice is trained with the local quadratic loss

$$\mathcal{L}_i(t, w_i) = \mathbb{E}\big[(w_i^\top [X, S_i(t)] - (Y(t) - Z))^2\big], \qquad w_i := \begin{bmatrix} u_i \\ v_i \end{bmatrix}.$$

Define the moment matrices

$$A_i(t) := \mathbb{E}[\tilde{X}_i(t)\tilde{X}_i(t)^\top], \qquad b_i(t) := \mathbb{E}[\tilde{X}_i(t)(Y(t) - Z)], \quad \tilde{X}_i(t) := [X, S_i(t)].$$

Then gradient flow is

$$\dot{w}_i(t) = -2(A_i(t)w_i(t) - b_i(t)). \tag{11}$$

## E.3. Gradient Flow of the Prediction

**Theorem E.2** (Exact decomposition). *The time derivative of the mean predictor $f(x;t) = \sum_m \beta_m(t)u_m(t)^\top x$ satisfies*

$$\frac{d}{dt}f(x;t) = \underbrace{\sum_{m=1}^{T-1} \beta_m(t)\,\dot{u}_m(t)^\top x}_{\text{Term 2a}} + \underbrace{\sum_{k=2}^{T-1} s_k(x,0;t)\frac{h\,\dot{v}_k(t)}{1 + hv_k(t)}}_{\text{Term 2b}}, \tag{12}$$

*where $s_k(x,0;t) = \sum_{m<k} \beta_m(t)u_m(t)^\top x$.*

*Equivalently,*

$$\frac{d}{dt}f(x;t) = \sum_{m=1}^{T-1} \beta_m(t)\,\dot{u}_m(t)^\top x + \sum_{m=1}^{T-2} \dot{\beta}_m(t)\, u_m(t)^\top x, \quad \dot{\beta}_m(t) = \beta_m(t)\sum_{k=m+1}^{T-1}\frac{h\,\dot{v}_k(t)}{1 + hv_k(t)}.$$

## E.4. Moving Targets: Explicit Gain Dynamics

Let

$$\Sigma := \mathbb{E}[XX^\top], \qquad b(t) := \mathbb{E}[XY(t)], \qquad q(t) := \mathbb{E}[Y(t)^2].$$

No distributional assumptions are made beyond existence of these moments.

**Lemma E.3** (Gain dynamics). *For each slice $k$,*

$$\dot{v}_k(t) = -2\Big((1 - \alpha_k)\,b(t)^\top u_k(t) + \big((1 - \alpha_k)^2 q(t) + \alpha_k^2\sigma_z^2\big)v_k(t) - (1 - \alpha_k)q(t) + \alpha_k\sigma_z^2\Big). \tag{13}$$

## E.5. Comparison with Monolithic Training

A monolithic linear predictor $f_{\text{mono}}(x;t) = w(t)^\top x$ trained by squared loss satisfies

$$\dot{w}(t) = -2(\Sigma w(t) - b(t)). \tag{14}$$

Thus its instantaneous movement is

$$\partial_t f_{\text{mono}}(x;t) = \dot{w}(t)^\top x,$$

and any adaptation requires direct motion of $w(t)$.

## E.6. Main Result

**Theorem E.4** (Feature-reweighting advantage under moving targets). *Assume $\Sigma \succ 0$ and finite second moments. Fix a time interval $[t_0, t_1]$ and suppose*

$$\dot{u}_m(t) = 0 \quad \text{for all } m \text{ and } t \in [t_0, t_1].$$

1. *(Monolithic) If $\partial_t f_{\mathrm{mono}}(\cdot; t) \neq 0$ on $[t_0, t_1]$, then necessarily $\dot{w}(t) \neq 0$.*

2. *(Flow matching) The Euler flow-matching predictor satisfies*

$$\partial_t f_{\mathrm{FM}}(x; t) = \Big( \sum_{m=1}^{T-2} \dot{\beta}_m(t) \, u_m \Big)^\top x,$$

   *with $\dot{\beta}_m(t)$ driven entirely by the gain dynamics $\{\dot{v}_k(t)\}$.*

3. *(Time-selective responsiveness) The sensitivity of $\dot{v}_k(t)$ to target drift satisfies*

$$\frac{\partial \dot{v}_k(t)}{\partial b(t)} = -2(1 - \alpha_k) u_k(t),$$

   *so later slices ($\alpha_k$ small) respond strongly to drift, while early slices respond weakly.*

*Hence, under moving targets, flow matching admits predictor motion via reweighting existing features (Term 2b), whereas monolithic training can adapt only by rewriting its feature vector.*

### E.6.1. INTERPRETATION

Although both models represent linear predictors, flow matching decomposes learning into two distinct channels: (i) feature learning through $\dot{u}_m$ (Term 2a), and (ii) feature reweighting through downstream gain updates $\dot{v}_k$ (Term 2b). Under moving targets, the gain dynamics are slice-dependent and explicitly modulated by the target moments, enabling rapid reallocation of importance among previously learned features without altering them. Monolithic training lacks this mechanism and must instead directly overwrite its parameter vector to track drift.

## E.7. Feature Reweighting via Downstream Gains

We formalize the notion of *feature reweighting* induced by Term 2b.

### E.7.1. DEFINITION (FEATURE REPRESENTATION)

At any training time $t$, define the collection of slice features

$$\mathcal{U}(t) := \{u_1(t), \dots, u_{T-1}(t)\} \subset \mathbb{R}^d,$$

and the corresponding effective predictor

$$f(x; t) = w_{\mathrm{eff}}(t)^\top x, \qquad w_{\mathrm{eff}}(t) := \sum_{m=1}^{T-1} \beta_m(t) \, u_m(t),$$

where

$$\beta_m(t) := h \prod_{j=m+1}^{T-1} (1 + h v_j(t)).$$

Thus the model represents a linear function as a weighted combination of fixed feature vectors $u_m(t)$.

### E.7.2. DEFINITION (FEATURE REWEIGHTING)

We say that the predictor undergoes *feature reweighting* on a time interval $I \subset \mathbb{R}_+$ if

$$\dot{u}_m(t) = 0 \quad \forall m, \ \forall t \in I, \qquad \text{but} \qquad \dot{\beta}_m(t) \neq 0 \text{ for some } m \text{ and } t \in I.$$

Equivalently, feature reweighting means that the predictor $f(\cdot; t)$ changes over time while the feature set $\mathcal{U}(t)$ remains fixed.

E.7.3. PROPOSITION (EXACT REWEIGHTING IDENTITY)

Under the Euler flow-matching dynamics,

$$\dot{w}_{\text{eff}}(t) = \sum_{m=1}^{T-1} \beta_m(t) \, \dot{u}_m(t) \; + \; \sum_{m=1}^{T-2} \dot{\beta}_m(t) \, u_m(t),$$

with

$$\dot{\beta}_m(t) = \beta_m(t) \sum_{k=m+1}^{T-1} \frac{h \, \dot{v}_k(t)}{1 + hv_k(t)}.$$

In particular, if $\dot{u}_m(t) = 0$ for all $m$ on an interval $I$, then

$$\dot{w}_{\text{eff}}(t) = \sum_{m=1}^{T-2} \dot{\beta}_m(t) \, u_m \quad \text{for all } t \in I,$$

so the predictor changes *purely* by reweighting existing features.

E.7.4. INTERPRETATION

The decomposition above shows that the Euler flow-matching model separates learning into two distinct mechanisms:

- **Feature learning (Term 2a):** updates of $u_m(t)$ change the feature directions available to the model.

- **Feature reweighting (Term 2b):** updates of downstream gains $\{v_k(t)\}$ modify the coefficients $\{\beta_m(t)\}$, thereby changing the relative importance of previously learned features without altering them.

Crucially, feature reweighting is mediated by downstream gains: changing a single $v_k(t)$ rescales *all* features injected at earlier slices $m < k$. This creates a hierarchical structure in which earlier features may be introduced tentatively and later amplified or suppressed depending on downstream dynamics.

This mechanism is absent in monolithic linear training. For a monolithic predictor $f_{\text{mono}}(x; t) = w(t)^\top x$, any change in the predictor necessarily requires $\dot{w}(t) \neq 0$, meaning that features themselves must be rewritten. In contrast, flow matching admits predictor adaptation through coefficient updates alone, enabling the model to track changes in the target by reallocating importance among existing features.

**Summary.** Feature reweighting is the ability of the flow-matching model to change its prediction by adjusting downstream gains, while keeping the feature directions fixed. Term 2b is exactly the mathematical expression of this mechanism.

### E.8. Why an Ensemble of Monolithic Networks Cannot Replicate Term 2b

In this section we formalize the statement that an ensemble of independently trained monolithic predictors cannot realize the *feature reweighting* mechanism provided by Term 2b in flow matching.

E.8.1. SETUP: MONOLITHIC ENSEMBLE

Let $(X, Y(t))$ be a time-indexed (moving-target) data process with finite second moments, and let $\pi_1, \ldots, \pi_K$ be fixed ensemble weights with $\pi_k \geq 0$ and $\sum_{k=1}^{K} \pi_k = 1$.

Each ensemble member is a monolithic linear predictor

$$f^{(k)}(x; t) := w_k(t)^\top x,$$

trained by gradient flow on the time-varying squared loss

$$\mathcal{L}^{(k)}(t, w) := \mathbb{E}\big[(w^\top X - Y(t))^2\big].$$

Then each member evolves according to

$$\dot{w}_k(t) = -2(\Sigma w_k(t) - b(t)), \qquad \Sigma := \mathbb{E}[XX^\top], \quad b(t) := \mathbb{E}[X\,Y(t)].$$

The ensemble predictor is defined as the weighted average

$$f_{\mathrm{ens}}(x;t) := \sum_{k=1}^{K} \pi_k f^{(k)}(x;t) = \sum_{k=1}^{K} \pi_k\, w_k(t)^\top x.$$

### E.8.2. THEOREM: NO FEATURE REWEIGHTING CHANNEL IN A FIXED-WEIGHT ENSEMBLE

**Theorem E.5** (Ensembles do not create a Term-2b-like reweighting mechanism). *Assume $\Sigma$ exists and is finite and that $b(t)$ exists for all $t$. Define the ensemble-averaged parameter*

$$\bar{w}(t) := \sum_{k=1}^{K} \pi_k w_k(t).$$

*Then:*

1. *(**Function-level collapse**) The ensemble predictor is exactly a single monolithic predictor:*

$$f_{\mathrm{ens}}(x;t) = \bar{w}(t)^\top x.$$

2. *(**Dynamics collapse**) The averaged parameter follows the same monolithic gradient flow:*

$$\dot{\bar{w}}(t) = -2\big(\Sigma\bar{w}(t) - b(t)\big).$$

3. *(**No reweighting without parameter motion**) For every $t$,*

$$\partial_t f_{\mathrm{ens}}(x;t) = \dot{\bar{w}}(t)^\top x = \sum_{k=1}^{K} \pi_k\, \dot{w}_k(t)^\top x.$$

   *In particular, if $\dot{w}_k(t) = 0$ for all $k$ on a time interval $I$, then $f_{\mathrm{ens}}(\cdot;t)$ is constant on $I$.*

   *Consequently, a fixed-weight ensemble cannot change its predictor by* reweighting previously learned features while keeping member parameters fixed*; any change in $f_{\mathrm{ens}}$ requires direct motion of at least one member parameter $w_k(t)$.*

*Proof.* (1) Linearity gives

$$f_{\mathrm{ens}}(x;t) = \sum_{k=1}^{K} \pi_k w_k(t)^\top x = \Big( \sum_{k=1}^{K} \pi_k w_k(t) \Big)^\top x = \bar{w}(t)^\top x.$$

(2) Differentiate $\bar{w}(t) = \sum_k \pi_k w_k(t)$ and substitute the gradient flows:

$$\dot{\bar{w}}(t) = \sum_{k=1}^{K} \pi_k \dot{w}_k(t) = \sum_{k=1}^{K} \pi_k\big( -2(\Sigma w_k(t) - b(t)) \big) = -2\Big(\Sigma \sum_{k=1}^{K} \pi_k w_k(t) - b(t) \sum_{k=1}^{K} \pi_k \Big),$$

which equals $-2(\Sigma\bar{w}(t) - b(t))$ since $\sum_k \pi_k = 1$.

(3) Differentiate $f_{\mathrm{ens}}(x;t) = \bar{w}(t)^\top x$ to obtain $\partial_t f_{\mathrm{ens}}(x;t) = \dot{\bar{w}}(t)^\top x$ and substitute $\dot{\bar{w}}(t) = \sum_k \pi_k \dot{w}_k(t)$. If $\dot{w}_k(t) = 0$ for all $k$ on $I$, then $\dot{\bar{w}}(t) = 0$ on $I$, hence $f_{\mathrm{ens}}(\cdot;t)$ is constant. $\square$

E.8.3. INTERPRETATION AND COMPARISON TO TERM 2B

Theorem E.5 shows that a fixed-weight ensemble does not introduce an additional adaptation channel: its mean prediction is governed by the same single-parameter monolithic dynamics.

By contrast, in the Euler flow-matching model the effective parameter admits the decomposition

$$
w_{\text{eff}}(t) = \sum_{m=1}^{T-1} \beta_m(t)\, u_m(t), \qquad \dot{w}_{\text{eff}}(t) = \sum_m \beta_m(t)\dot{u}_m(t) + \sum_m \dot{\beta}_m(t)u_m(t),
$$

so the predictor can move even when $\dot{u}_m(t) = 0$ via the coefficient dynamics $\dot{\beta}_m(t)$ induced by downstream gains (Term 2b). The ensemble lacks any analog of coefficients $\{\beta_m(t)\}$ that can evolve independently of the base feature parameters, hence it cannot realize feature reweighting in this sense.

**Remark.** If one allows the ensemble weights $\pi_k$ themselves to vary with $t$ or to depend on $x$ (a learned gating network), then additional mechanisms become possible. However, such a model is no longer an ensemble of *independent monolithic networks with fixed averaging*; it introduces an explicit mixer/gate, which is precisely the kind of additional structure that Term 2b provides in the flow-matching construction.

## F. Strengthened Linear Analysis (Theorem 6.1)

Theorem 6.1 establishes that flow-matching critics can adapt purely through gain dynamics when feature directions are frozen. We now strengthen this result by removing the assumption that features are fixed, and instead quantify how much *feature motion* is required to track a non-stationary target.

Our goal is to show that even when $\dot{u}_t(m) \neq 0$, flow-matching critics can realize the same predictor change with *less feature movement* than monolithic critics, because part of the change can be absorbed through gain-induced reweighting.

**Rescaled representation.** To isolate the effect of gain dynamics, we rewrite Eq. Eq. 5 by absorbing the step sizes $\alpha_t$ into the features:

$$
\bar{u}_t(m) := \alpha_t u_t(m), \qquad \bar{\beta}_t(m) := \prod_{k=t+1}^{T-1} \big(1 + \alpha_k v_k(m)\big). \tag{15}
$$

Then the effective weight becomes

$$
w_{\text{FM}}(m) = \sum_{t=1}^{T-1} \bar{\beta}_t(m)\, \bar{u}_t(m) = \bar{U}(m)\bar{\beta}(m), \tag{16}
$$

where $\bar{U}(m) = [\bar{u}_1, \dots, \bar{u}_{T-1}]$.

We now compare how much feature motion is *minimally required* to realize a desired predictor change.

**Theorem F.1** (**Flow matching reduces required feature motion under target drift**). *Fix a training step $m$, and let $g \in \mathbb{R}^d$ denote a desired change in the effective weight:*

$$\dot{w}(m) = g.$$

*Define the minimum feature motion for the flow model as*

$$\mathcal{M}_{\mathrm{FM}}(g) := \inf_{\dot{U}, \dot{\bar{\beta}}: \bar{U}\dot{\bar{\beta}} + \dot{U}\bar{\beta} = g} \|\dot{U}\|_F,$$

*and for a monolithic model*

$$\mathcal{M}_{\mathrm{mono}}(g) := \inf_{\dot{U}: \dot{U}\mathbf{1} = g} \|\dot{U}\|_F.$$

*Then*

$$\mathcal{M}_{\mathrm{FM}}(g) = \frac{\mathrm{dist}(g, \mathrm{span}(\bar{U}))}{\|\bar{\beta}\|_2},$$

$$\mathcal{M}_{\mathrm{mono}}(g) = \frac{\|g\|_2}{\sqrt{T-1}}.$$

*Consequently,*

$$\frac{\mathcal{M}_{\mathrm{FM}}(g)}{\mathcal{M}_{\mathrm{mono}}(g)} = \frac{\sqrt{T-1}}{\|\bar{\beta}\|_2} \cdot \frac{\mathrm{dist}(g, \mathrm{span}(\bar{U}))}{\|g\|_2}.$$

*In particular, letting $\theta$ denote the angle between $g$ and $\mathrm{span}(\bar{U})$,*

$$\frac{\mathcal{M}_{\mathrm{FM}}(g)}{\mathcal{M}_{\mathrm{mono}}(g)} = \frac{\sqrt{T-1}}{\|\bar{\beta}\|_2} \sin\theta.$$

**Proof sketch.** The flow predictor satisfies

$$w_{\mathrm{FM}} = \bar{U}\bar{\beta} \quad \Rightarrow \quad \dot{w}_{\mathrm{FM}} = \bar{U}\dot{\bar{\beta}} + \dot{U}\bar{\beta}.$$

Thus predictor change decomposes into: (i) reweighting existing features ($\bar{U}\dot{\bar{\beta}}$), and (ii) feature motion ($\dot{U}\bar{\beta}$).

Fix $\dot{\bar{\beta}}$ and define the residual $r = g - \bar{U}\dot{\bar{\beta}}$. The minimum-norm solution to $\dot{U}\bar{\beta} = r$ is a rank-one update with norm $\|r\|_2/\|\bar{\beta}\|_2$. Minimizing over $\dot{\bar{\beta}}$ yields

$$\mathcal{M}_{\mathrm{FM}}(g) = \frac{\mathrm{dist}(g, \mathrm{span}(\bar{U}))}{\|\bar{\beta}\|_2}.$$

For the monolithic model, $\dot{w} = \dot{U}\mathbf{1}$, and the minimum-norm solution gives

$$\mathcal{M}_{\mathrm{mono}}(g) = \frac{\|g\|_2}{\sqrt{T-1}}.$$

Taking the ratio yields the result. $\qquad\square$

**Discussion.** Theorem F.1 formalizes a stronger notion of plasticity:

- **Flow critics separate adaptation channels.** Predictor change decomposes into reweighting ($\bar{U}\dot{\bar{\beta}}$) and feature updates ($\dot{U}$). Monolithic critics lack this separation.

- **Feature motion depends only on unexplained drift.** Flow critics only need to move features for the component of $g$ orthogonal to $\text{span}(\bar{U})$.

- **Strict advantage beyond the frozen-feature case.** Flow matching outperforms monolithic models whenever

$$\sin \theta < \frac{\|\bar{\beta}\|_2}{\sqrt{T-1}},$$

i.e., whenever target drift is sufficiently aligned with existing features.

- **Natural regime.** When $\|\bar{\beta}\|_2/\sqrt{T-1} = \Theta(1)$ (which holds when gain-induced coefficients remain order-one), the ratio simplifies to

$$\frac{\mathcal{M}_{\text{FM}}}{\mathcal{M}_{\text{mono}}} = \Theta(\sin \theta).$$

Thus the benefit of flow matching is governed primarily by geometric alignment: the more aligned the future target drift is with previously learned features, the less feature rewriting is required.

Overall, this strengthens Theorem 6.1 by showing that the advantage of flow matching is not limited to the degenerate frozen-feature regime, but persists quantitatively whenever target drift can be partially explained by reweighting existing features.

## G. Discussion and Perspectives on Future Work

This work explains the effectiveness of flow-matching critics in off-policy reinforcement learning by identifying dense, trajectory-level supervision as the key mechanism. Rather than gains from distributional modeling or expressivity, flow-matching critics learn a velocity field jointly with an integration procedure, enabling test-time recovery and preserving representation plasticity under non-stationary TD targets. In contrast to monolithic critics, this coupling between training and iterative computation allows robust adaptation to noise and target drift, highlighting dense intermediate supervision as a powerful inductive bias for stabilizing TD learning.

Our findings open several directions for future work. On the practical side, it would be natural to explore alternative design choices for training flow-matching critics, such as representing velocities with higher-dimensional vectors rather than scalars, as well as to evaluate their effectiveness in settings that place stronger demands on plasticity, including continual learning with shifting task distributions. From a theoretical perspective, extending our analysis beyond the linear setting to nonlinear function approximation remains an important open problem. More broadly, our results point to interesting connections between plasticity under non-stationarity, flow matching, and test-time computation. While we study these interactions in the context of TD learning, the underlying principles governing flow-matching dynamics may extend to other domains, such as time-series modeling, suggesting a broader set of applications and theoretical questions for future investigation.

