# OpenReview forum: "What Does Flow-Matching Bring to TD-Learning?"
_ICML.cc/2026/Conference — ICML 2026 regular_

### Official Review · Reviewer_5WE4 · 2026-02-19

**Soundness:** 3
**Presentation:** 2
**Significance:** 2
**Originality:** 2
**Overall Recommendation:** 5
**Confidence:** 3

**Summary:**

This paper shows that the performances of flow-matching networks are not explained by distributional RL and explicitly modeling return distributions often degrades performance. Flow-matching Q-functions, during both training and inference time, couple a learned velocity field. They have increased performances due to test-time recovery from imperfect intermediate estimates where more integration steps are performed. Beyond test-time recovery, training with the integration procedure induces more plastic representations. The author formalizes these effects and validates them empirically, showing that flow-matching critics outperform monolithic critics in terms of both performance and sample efficiency.

**Compliance With Llm Reviewing Policy:**

Affirmed.

**Final Justification:**

The author response has addressed my concerns adequately. Therefore, I have raised my score to 5.

**Key Questions For Authors:**

See Weakness

**Strengths And Weaknesses:**

***Strength***

1. The paper coined the notions of test-time recovery and the preservation of plasticity of flow-matching networks, giving insights into flow-matching critics and monolithic critics.

2. The main argument that distributional RL is an insufficient explanation is easy to agree with.

3. The results presented in the paper empirically show that flow-matching critics tolerate higher noise or interventions and learn more isotropic
features.

***Weakness***

1. The theory in a linear setting is too simplified to explain behaviors under nonlinear deep RL.

2. The concept of flow-matching networks is not new. I recognize the contribution of this work but it is also limited.

---

> ### Author Rebuttal · Authors · 2026-03-31
>
> Thanks for your feedback! To address your concerns regarding theory, we now add an extension of our theoretical result that extends the theoretical statement to more general linear settings (https://anonymous.4open.science/r/rebuttal-C117/new_revised_strengthened_stds.pdf, Section 2). We also outline possible expansions of our theory in the non-linear setting below and would be grateful if you are willing to advise us on what version we should use to expand our theory to the non-linear setting.
>
>
> > The theory in a linear setting is too simplified to explain behaviors under nonlinear deep RL.
>
> We agree that the linear setting is simplified and does not fully capture the behavior of nonlinear deep RL. Our intention was not to provide a complete account of neural-network TD learning, but rather to use a tractable setting to isolate the specific mechanism we study: how flow-matching critics can adapt to changing TD targets through reweighting along the integration procedure, in contrast to monolithic critics that must modify features directly. We also agree that extending the analysis beyond the linear setting would strengthen the paper. To this end, we see two natural directions we would be happy to incorporate in the revision:
>
> 1. In the Neural Tangent Kernel regime, neural networks are linearized around their parameters and operate with effectively fixed features, with learning occurring through coefficient updates. This setting provides a principled bridge between linear models and deep networks. Our linear result translates naturally to this regime, since the key distinction we identify (i.e. adaptation via reweighting of existing features vs. modification of features) remains meaningful. In particular, flow-matching critics induce a structured reweighting through the integration dynamics, allowing adaptation under fixed features, whereas monolithic critics lack this mechanism under the same kernel. This would provide a more realistic theoretical setting while preserving the core mechanism we analyze.
>
> 2. Alternatively, we can extend the analysis to a local nonlinear regime around a trained solution, in the spirit of prior work analyzing TD learning dynamics (e.g. DR3 [1]). This would allow us to study how changes in TD targets affect representations and gradients, and to formally characterize how flow-matching critics absorb target shifts through integration-induced reweighting rather than feature updates, directly connecting to plasticity.
>
> We would be happy to include an extension along either direction, and would appreciate the reviewer’s guidance on which would be most valuable. Finally, we would like to emphasize that the primary support for our claims comes from the controlled empirical results (staleness injection, noisy TD targets, feature freezing, feature norms, and high-UTD scaling), which consistently demonstrate the proposed mechanisms in nonlinear settings. We look forward to your opinion on the expansion of our theoretical results.
>
> [1] DR3: Value-Based Deep RL Requires Explicit Regularization, Kumar et. al 2021.
>
> > The concept of flow-matching networks is not new. I recognize the contribution of this work but it is also limited.
>
> We agree that flow-matching models are not new, and we do not claim to introduce this class of methods. Our work builds directly on prior papers that apply flow matching to TD learning and demonstrate empirical improvements over monolithic critics. **Our contribution is to study why these improvements arise**. In particular, we show that the gains are not due to distributional value modeling (which has been the motivating factor for a large chunk of papers on flow-matching for value estimation!), and instead identify two mechanisms (test-time recovery and improved feature plasticity) that emerge from the flow-matching TD objective. Building on this understanding, we also identify settings where these properties are especially beneficial (e.g. high-UTD training). To our knowledge, prior work has not provided a mechanistic explanation of the advantages of flow-matching critics in TD learning, and we will clarify this positioning in the manuscript.
>
> Thank you again for the thoughtful feedback and suggestions. We hope that the clarifications and proposed extensions address the concerns regarding the theoretical component. We would greatly appreciate any additional guidance on how to further strengthen this aspect of the paper. Overall, we hope that our responses and the planned revisions help clarify the contribution and motivate reconsideration of the score.

---

> > ### Author Rebuttal · Reviewer_5WE4 · 2026-04-01
> >
> > Thank you for the rebuttal. The answers are informative and have addressed my concerns adequately. Therefore, I would like to raise my score to 5.

---

### Official Review · Reviewer_iTCC · 2026-02-27

**Soundness:** 3
**Presentation:** 2
**Significance:** 4
**Originality:** 4
**Overall Recommendation:** 4
**Confidence:** 4

**Summary:**

This paper addresses a core question in flow-model-based TD-learning: what is the true source of advantage when using a flow model as the critic. The authors begin by dispelling a common misconception — the benefit does not stem from modeling the return distribution, but rather from the natural iterative integration mechanism inherent to flow models during both training and inference. Building on this, they analyze the advantages of the mechanism from two perspectives. First, it introduces test-time recovery (TTR) at inference: even when individual integration steps incur errors, the overall quality remains robust, as the dense supervision signal during training endows each step with error-correcting capability, causing per-step errors to attenuate over steps. Second, the author claims that iterative integration promotes greater representational flexibility and improved network plasticity. Both claims are supported experimentally and theoretically. Overall, the insights are non-trivial and the core thesis is compelling, though the paper suffers from disorganized exposition and inconsistent notation in several places.

**Compliance With Llm Reviewing Policy:**

Affirmed.

**Final Justification:**

My major concerns have been addressed, but I still think the authors need to improve the presentation and writing of this paper — particularly in the theoretical sections — to make them clearer. Therefore I will maintain my score. Overall, this is a good paper.

**Key Questions For Authors:**

1. Is the linear Euler recursion defined in Appendix E actually used in the experiments? If in practice the velocity network simply takes $s, a, z(t)$ as joint inputs, how large is the gap between the theoretical analysis and the actual implementation? Under the theoretical formulation, some parameters should be responsible for producing representations from $s$ and $a$, while others serve as gain parameters. This distinction matters: a small layer parameter norm does not necessarily indicate greater representational plasticity, since it is difficult to disentangle which parameters correspond to the representation and which to the gain.
2. Why is there such a large discrepancy between the SARSA and TD-learning results in Figures 3 and 9? If I understand correctly, both use non-stationary targets. Yet the feature norm under SARSA appears to grow continuously --- could the authors clarify this?
3. I would like to raise a point regarding the section "Experiment: Intermediate velocity supervision is crucial." Based on [1], I believe the key difference between "Predict Final" and "Predict Velocity" lies in the fact that the "Predict Final" objective in [1] amplifies the loss as $t \rightarrow 1$, due to a $\frac{1}{1-t}$ factor in the training objective. This design likely encourages the learned velocity to be more precise in the later stages of generation, yielding final outputs closer to the target. However, in TD-learning, the target itself may be inaccurate, and placing excessive emphasis on matching the final output to the target could amplify erroneous supervision signals. Put differently, what matters for a flow model in TD-learning is not the final output, but the iterative integration process itself --- a fundamental difference from image generation. Could the authors comment on this perspective?

[1]: Li, T. and He, K. Back to basics: Let denoising generative models denoise. arXiv preprint arXiv:2511.13720, 2025

**Limitations:**

yes

**Strengths And Weaknesses:**

Strengths:
1. The studied problem is important. Using a flow model as the critic is a recently emerging practice. Since flow models are conventionally used for distribution modeling, it is natural to attribute the advantage of flow critics to their superior capacity for modeling the return distribution. The authors challenge this assumption with substantial evidence, arguing that the true advantage lies instead in the model's ability to more fully exploit the information in supervision signals under high-stochasticity environments, and to perform more robust inference.
2. The main claims are well-supported. Both theoretical and empirical evidence are provided throughout. The test-time recovery analysis is particularly compelling, offering valuable intuition as well as formal justification.
3. The work offers practical insights. The authors not only demonstrate that modeling the return distribution is not a necessary condition for an effective flow critic, but also highlight the importance of providing direct supervision on the velocity field within the TD-learning framework.

Weaknesses:
1. Some experimental results are insufficient. For instance, the experiments in Figures 2 and 3 cover only a limited number of environments, and the paper as a whole uses only a small subset of OG-Bench. Furthermore, including baselines that also adopt a flow critic but differ from floq would make the conclusions more convincing.
2. The presentation is at times disorganized. For example, in Section 6.1, $t$ appears to refer to training time, which is easily confused with the time variable in flow matching. Additionally, the definitions of $u$ and $v$ are only listed in the appendix, where they are revealed to be parameters of the linear Euler recursion --- this should be made explicit in the main text, as the current presentation significantly hinders readability.
3. Theorem 2 is relatively weak. It only shows that, even when the feature $u_m$ remains fixed, changes in the output can be fully accounted for by changes in the gain parameter $v_k$. However, in practice, output changes may well be dominated by changes in $u_m$. Moreover, the proof itself is straightforward --- it amounts to expanding the gradients of the expected output $f(x;t)$ with respect to $v_k$ and $u_m$, and invoking the assumption that the gradient with respect to $u_m$ is zero. The result is therefore somewhat naive.

---

> ### Author Rebuttal · Authors · 2026-03-31
>
> Thank you for the feedback. We include a strengthened analysis of Theorem 2 in the attached PDF (https://anonymous.4open.science/r/rebuttal-C117/new_revised_strengthened_stds.pdf, Section 2). We address the concerns below. If resolved, we would appreciate reconsideration of the score.
>
>
> > The experiments in Figures 2 and 3 cover only a limited number of environments, and the paper as a whole uses only a small subset of OG-Bench.
>
> Across experiments, we used distinct but representative subsets of the 6-7 hard OG-Bench environments (out of 10 total envs), where monolithic critics typically perform only moderately per results in Agrawalla et al. 2025.
> For target noise (Figure 2), we chose three hard environments where floq and FQL have similar zero-noise performance, so differences under added noise more cleanly reflect robustness over other confounding factors. For the feature norms and plasticity results (Figures 3, 4, 8), we used 4 hard environments spanning different task categories. For RLPD (Figure 6), we selected the four hard long-horizon environments, where a higher discount factor is needed to stress-test the method. We also already include two additional environments for the feature norms analysis in Figure 8, bringing that total to four.
> That said, we agree that broader coverage would strengthen the evaluation, and will add at least 2 more environments for each experiment in the final.
>
> > Including baselines that also use a flow critic but differ from floq would make the conclusions more  convincing.
>
> We agree alternative flow-critic baselines would strengthen the conclusions. Nearly all prior work we found uses distributional RL, such as Dong et al. (2025) and Espinosa-Dice et al. (2025). As we show in Section 4, modeling the return distribution does not explain the gains in our setting and can hurt performance. We are not aware of prior non-distributional flow-matching critics. We also considered adapting distributional methods, but did not find a clean way to do so.
>
> >  Section 6.1; Strength of Theorem 2
>
> Thanks a lot for the feedback! We agree that Section 6.1 can be made clearer and we do so in the attached PDF (https://anonymous.4open.science/r/rebuttal-C117/new_revised_strengthened_stds.pdf, Section 1). We also present a stronger version of Theorem 2 in this document (Section 2). This version removes the assumption that features $u_t$ do not move and writes the general condition of smaller magnitude feature updates in flow-matching critics when $u_t$ and $v_t$ are both allowed to change. Intuitively, this is the case when future target drift is roughly aligned with previously learned features. **Please let us know if this result addresses your concerns.** We will also include these in the final.
>
> > Theory-practice gap for Euler recursion.
>
> We do not use the explicit linear Euler recursion in practice, and will note this as a limitation. Our goal in Section 6.1 is not to model the practical implementation exactly, but to provide an analytically tractable setting that separates feature directions from gain-based reweighting induced by integration to build intuition. The experiments show behavior consistent with this picture, e.g., robustness to freezing features (Figure 4).
>
> > SARSA vs TD-learning gap
>
> While both floq(SARSA) and floq(TD) involve non-stationary targets, in practice **SARSA targets are substantially more stationary**, especially in later stages of training. We verify this empirically on cube-double by measuring the magnitude of change in targets (per training step) on a held-out dataset:
>
> |                              | floq (TD) | floq (SARSA) |
> |-------------------------------------|-----------|--------------|
> | **\|Change in target\| (per training step)** | 0.5       | 0.1          |
>
> When targets are stationary, we hypothesize that a flow-matching or monolithic network will behave similarly when training for longer, resulting in different conclusions.
>
> > Predict final vs predict velocity
>
> This is a good point. “Predict Final” places significantly more emphasis on matching the final output, which is beneficial in image generation where the target is accurate.
> However, in TD learning, the target is **bootstrapped and inherently noisy/non-stationary**. In this setting, emphasizing precise matching of the final output can amplify errors in the supervision signal. In effect, the “Predict Final” objective pushes the model to represent the full (and shifting) TD target at every point along the trajectory, which makes it behave similarly to a monolithic critic.
> Following the reference, we also implemented a variant of “Predict Final” with explicit $1/(1-t)$ loss weighting. We find that it still performs significantly worse than velocity supervision:
>
>
> | | “Predict Final” ($1/(1-t)$ loss weighting) | “Predict Velocity” |
> |---------------|-------------------------------------|--------------------|
> | Antmaze-Giant | 45 ±10 | **86**±4 |
> | Cube-Double | 40 ±20 | **72**±15 |

---

> > ### Author Rebuttal · Reviewer_iTCC · 2026-04-01
> >
> > Concerning the "SARSA vs. TD-learning gap", I have a point of confusion. In terms of target stationarity, one would naturally expect the ordering MC (fully stationary) > SARSA (relatively stationary) > TD-learning (unstationary). However, as shown in Figure 3(b), the observed ordering of penultimate hidden layer feature norms is SARSA > MC > TD-learning, which does not align with this expectation. While I acknowledge that floq consistently yields lower feature norms than FQL across all settings, this discrepancy raises the question of whether target non-stationarity is indeed the primary driver of feature norm growth. I would ask the authors to provide further clarification on this point.

---

> > > ### Author Response · Authors · 2026-04-02
> > >
> > > Thanks for this question and for engaging with our work. We appreciate the opportunity to clarify two points.
> > >
> > > 1. First, the claim we make in the paper is about comparing feature-norm trends between flow-matching and monolithic critics while holding the learning algorithm fixed. Concretely, we compare TD (floq) with TD (FQL), SARSA (floq) with SARSA (FQL), and MC (floq) with MC (FQL). In both the paper and the rebuttal, we do not intend to claim that absolute feature norms should be compared across different learning algorithms. These norms depend on the exact training targets, the loss function, and whether Bellman bootstrapping is used, so they can vary substantially across environments and objectives. For this reason, feature-norm trends are most meaningful when the learning algorithm is held fixed, which is the setting we focus on.
> > >
> > > This point is also consistent with prior work. For example, Kumar et al. (2021, DR3) show that, for standard monolithic critics in their setting, SARSA yields lower feature norms than MC regression, whereas in our setting we observe the opposite trend. This suggests that comparisons of feature norm values across algorithms are themselves can be environment dependent. We will clarify this more explicitly in the paper.
> > >
> > > That said, we believe the gap between SARSA and MC regression is likely explained by the use of bootstrapping and differences in the regression objective. Even though MC regression is stationary, TD and SARSA regress to targets whose scale and distribution evolve over training, making absolute feature norms difficult to compare directly with MC, which regresses on to a high value static target. In contrast, TD and SARSA are somewhat more comparable to each other, since they use similar loss functions and both rely on bootstrapped targets, differing primarily in the degree of target non-stationarity.
> > >
> > > 2. Second, our claim is not that there is a direct causal link between non-stationarity and feature norms. We use feature norms because prior work has treated them as one indicator of representational pathology under bootstrapping. However, our main evidence for improved plasticity is not the feature-norm plot itself, but the intervention experiment in Figure 4, where we freeze features during training. In that experiment, floq critics recover from the intervention, whereas monolithic critics do not. We view this as the stronger piece of evidence, because it more directly shows that flow-matching critics learn representations that remain adaptable throughout training, rather than relying on feature norms alone as a proxy. We will make this point clearer in the paper.

---

### Official Review · Reviewer_6wck · 2026-03-08

**Soundness:** 2
**Presentation:** 2
**Significance:** 3
**Originality:** 3
**Overall Recommendation:** 5
**Confidence:** 3

**Summary:**

This paper aims to provide a mechanistic explanation of the performance gains provided by flow matching in the context of TD-learning in RL. The paper makes three major claims: 1) the benefits of flow-matching critics are not due to learning the full Q-value distribution. 2) Flow-matching critics exhibit test-time recovery, where intermediate errors in the Q-value prediction can be corrected by later integration steps. 3) Flow-matching critics learn more plastic features, as the integration procedure acts as a buffer between the TD targets and the learned features. The paper provides both theoretical and empirical evidence to support these claims.

**Compliance With Llm Reviewing Policy:**

Affirmed.

**Final Justification:**

I provided an initial rating of 5, with only clarifying questions, which the authors have adequately addressed. I thus keep my rating.

**Key Questions For Authors:**

- The ability of flow-matching critics to perform test-time recovery seems to hinge on the accuracy of the velocity fields predicted late in the integration process (for large $t \in [0, 1]$).  If it is the other way round, meaning that your early velocity fields are accurate, but later ones are inaccurate, would flow-matching critics predict wrong Q-values for many state-action pairs? Then whether this integration procedure is a good thing really seems to depend on which case you have.
- In Section 6.2, can you remind me of the difference between TD-learning and SARSA? I find the naming a bit confusing cause SARSA is also TD-learning.

**Limitations:**

yes

**Strengths And Weaknesses:**

Strengths:
- The paper improves understanding of the benefits of flow-matching in TD learning, by providing a wide variety of theoretical and empirical evidence.
- The paper is in general technically sound (I did not study Section 6.1/Appendix in detail, but others look good).
- The paper answers an important question, and will likely be relevant for future research.

Weaknesses:
1. In Section 5.2, the authors provide empirical evidence for the claim that flow-matching performs test time recovery, which means that the intermediate errors during the integration can be corrected by later integration steps. I have two questions about the results.
	- In the first experiment, you demonstrate the effect of test-time recovery by using an earlier checkpoint, instead of the current network, to generate the velocity fields for the early integration steps. If flow-matching exhibits test-time recovery, these staleness/error in the early velocity fields can be corrected by the later velocity fields generated by the current neural network. I have two questions here: (1) For the stale velocity fields, you use the checkpoint at $T=250,000$ gradient steps. Has the flow-matching Q-network converged at $T$ steps? To what extent are these stale velocity fields similar to the current ones? (2) You claim that flow-matching Q exhibits test-time recovery by showing (in Table 2) that when using stale velocity fields for the first 25-50% of the integration, the resulting success rate is even higher than no stale velocity fields at all (0%). This does not make sense to me. Shouldn't 0% be the best? If there is no error to correct from to begin with, shouldn't it be better than having error? I do not see an explanation of this phenomenon.
2. Details on statistical significance are missing. Number of seeds is not mentioned. Conf. int. or standard error not provided for success rates in tables. What shaded regions represent in figures is not explained.
3. Section 6.1 is very hard to parse. Improved clarity and self-containedness is appreciated.

---

> ### Author Rebuttal · Authors · 2026-03-31
>
> Thank you for your thoughtful review! We are glad you find the paper important for future research. Below, we address your questions on Section 5.2, present an improved presentation of Section 6.1, and clarify details. We hope these responses resolve your concerns and would greatly appreciate reconsideration of the score.
>
> > For the stale velocity field, you use the checkpoint at T=250K gradient steps…
>
> To quantify this, we measure how much the mixed integration output deviates from using fully stale or fully current velocity fields:
>
> $\delta_{stale}(\kappa) = \frac{|E_z[Q_{mixed}] - E_z[Q_{stale}]|}{Std_z[Q_{stale}]}, \delta_{current}(\kappa) = \frac{|E_z[Q_{mixed}] - E_z[Q_{current}]|}{Std_z[Q_{current}]}$
>
> In the table below, we compare the mixed integration output against fully stale and fully current velocity fields using the normalized deviations defined above. On Cube-Double, even at $\kappa=25-50$ percent, we find a. $\delta_{stale}\approx 10$ and b. $\delta_{current} \ll \delta_{stale}$, which show a.) **substantial evolution of velocity field beyond T=250K** and b.) **later integration steps from the current velocity field can correct errors introduced by earlier (stale) integration steps** respectively. We will include these results in the revision.
>
> | Staleness $\kappa$ (%) | $\delta_{stale}(\kappa)$ | $\delta_{current}(\kappa)$ |
> |--------------------------------|------------------------------|------------------|
> | 0                              | 16                           | 0               |
> | 25                             | 10                           | 0.5               |
> | 50                             | 10                           | 1.2               |
> | 75                             | 3                            | 1.5              |
> | 100 (fully stale)              | 0                            | 2               |
>
> > Why can $\kappa>0$ outperform $\kappa=0$?
>
> Our additional analysis suggests this is due to lower Q-value variance under mixed integration, which can yield more stable policy extraction. We verify this by measuring the success rate (left) and variance of the integration output over noise $z$ (right):
>
> | $\kappa$ (%) | AntSoccer        | HMMaze       | Cube         |
> | ------------ | ---------------- | ------------ | ------------ |
> | 0            | 45, 0.35         | **98, 0.35** | 72, 1.24     |
> | 25           | **50**, **0.13** | 43, 0.47     | **84, 0.37** |
> | 50           | 47, 0.16         | 63, 0.49     | 82, 0.37     |
> | 75           | 46, 0.27         | 62, 0.75     | 58, 0.65     |
> | 100          | 35, 0.63         | 39, 1.15     | 64, 1.64     |
>
> We observe that lower variance often correlates with better performance at intermediate $\kappa$. While the effect is environment-dependent, mixed integration can in some cases produce more stable Q-values and better performance.
>
> > Section 6.1 is very hard to parse…
>
> We agree Section 6.1 should be clearer and have revised it (https://anonymous.4open.science/r/rebuttal-C117/new_revised_strengthened_stds.pdf, Section 1). In short, we study a minimal linear setting where monolithic and flow-matching critics represent the same function class but differ in how they adapt to non-stationary TD targets. The key point is that a monolithic model must update all weights to track target drift, whereas a flow-matching model can also adapt by reweighting existing features through the gains, reducing feature rewriting. We also provide a stronger version of the theorem in the attached PDF (Section 2).
>
> > Details on statistical significance are missing…
>
> All experiments use 3 seeds; the attached PDF (https://anonymous.4open.science/r/rebuttal-C117/new_revised_strengthened_stds.pdf, Section 3) reports std. errors, and we will clarify in the paper that the error bars and shaded regions denote standard errors across seeds.
>
> > The ability to perform TTR hinges on the accuracy for large t \in [0,1]...
>
> We agree that inaccurate late-timestep velocities can corrupt the final Q-value, since later steps dominate the integration. To test this, we vary the sampling of t during training: Skew-Earlier ($t = 1-u, u \sim Beta(4,1)$), Default ($t \sim Unif[0,1]$), and Skew-Later ($t \sim Beta(4,1)$).
>
> | Environment | Skew-Earlier | Default | Skew-Later |
> |--------------|-----------|--------|------|
> | AntSoccer            |35 ±6      | **49** ±10   | 45 ± 8 |
> | Cube-Double           | 0 ±0  | 72 ±15   | **75** ±10|
>
> Skew-Earlier degrades performance substantially, confirming that late-step accuracy is critical for ensuring TTR operates normally. Skew-Later performs comparably to Default, suggesting that accuracy of late-stage velocities in Default is sufficient for TTR to work.
>
> > SARSA vs TD-learning
>
> By SARSA, we mean that the target uses dataset actions $a’$ at the next state $s’$, whereas TD-learning uses $a' \sim \pi(\cdot | s')$ from the current policy. Both still optimize temporal-difference error. We will clarify this in the paper.

---

> > ### Author Rebuttal · Reviewer_6wck · 2026-04-02
> >
> > Thanks for the detailed response. I'm happy to keep my positive rating.
> >
> > One final lingering question: why can mixed integration lower Q-value variance sometimes?

---

> > > ### Author Response · Authors · 2026-04-02
> > >
> > > This is a great question. Our current hypothesis is that this effect arises from target non-stationarity in TD learning. The flow critic is trained using supervision defined by interpolants between noise and Q-values produced by a stale copy of the Q-network. In other words, the supervision is placed on points obtained by linearly interpolating between the initial noise and target values coming from this lagged target network. At inference time, however, we estimate quantities such as the variance by integrating the current critic's velocity field.
> > >
> > > Because the target and online velocity networks are not perfectly aligned, the trajectory obtained by integrating the current velocity field can drift away from the target-network interpolation path on which training supervision was provided. As a result, the model may be evaluated along integration trajectories that differ from those it was most directly trained on and might result in a higher variance in the Q-values. This is analogous to the mismatch between teacher forcing and sampling in autoregressive models: during training, the model is supervised on a fixed trajectory, while at inference time it must generate its own trajectory by integrating the learned velocity field.
> > >
> > > In contrast, using a stale vector field during the early integration steps is more likely to keep the trajectory closer to interpolants that were well supervised during training, since those interpolants are themselves induced by target Q-values from a lagged network. This likely reduces the train-test mismatch to some extent and hence reduces variance of the Q-value predictions.

---

### Decision · Program_Chairs · 2026-04-30

**Decision:**

Accept (regular)

**Comment:**

This manuscript's principal aspect pertains to explaining why flow-matching critics improve TD learning beyond simple empirical gains. The work examines a significant challenge in reinforcement learning, namely instability and loss of plasticity under non-stationary TD targets, and argues convincingly that the benefit of flow matching comes not mainly from distributional value modeling, but from the coupling of velocity prediction and integration, which yields test-time recovery and more plastic representations. I find the submission strong overall: it rules out a natural alternative explanation, supports its claims with focused ablations and theoretical analysis, and demonstrates meaningful practical gains in high-UTD settings. While some theoretical assumptions may be stronger than what can currently be verified in full nonlinear settings, and the mechanism is not equally strong on every task, the paper provides a clear, original, and useful contribution to understanding flow-based critics. Overall, I support accept.